# MindFlayer: Efficient Asynchronous Parallel SGD in the Presence of Heterogeneous and Random Worker Compute Times

## Abstract

We study the problem of minimizing the expectation of smooth nonconvex functions with the help of several parallel workers whose role is to compute stochastic gradients. In particular, we focus on the challenging situation where the workers' compute times are arbitrarily heterogeneous and random. In the simpler regime characterized by arbitrarily heterogeneous but deterministic compute times, Tyurin & Richtárik (2024) recently proposed the first optimal asynchronous SGD method, called Rennala SGD, in terms of a novel complexity notion called time complexity. The starting point of our work is the observation that Rennala SGD can have arbitrarily bad performance in the presence of random compute times – a setting it was not designed to handle. To advance our understanding of stochastic optimization in this challenging regime, we propose a new asynchronous SGD method, for which we coin the name MindFlayer SGD. Our theory and empirical results demonstrate the superiority of MindFlayer SGD over existing baselines, including Rennala SGD, in cases when the noise is heavy tailed.

## 1 Introduction

We address the nonconvex optimization problem:

$$\min_{x \in \mathbb{R}^d} \left\{ f(x) := \mathbb{E}_{\xi \sim \mathcal{D}} \left[ f(x; \xi) \right] \right\}, \tag{1}$$

where $f : \mathbb{R}^d \times \mathbb{S} \to \mathbb{R}$, and $\xi$ is a random variable with some distribution $\mathcal{D}$ on $\mathbb{S}$. In the context of machine learning, $\mathbb{S}$ could represent the space of all possible data, $\mathcal{D}$ denotes the distribution of the training dataset, and $f(\cdot, \xi)$ denotes the loss of a data sample $\xi$.

The function $f$ is assumed to be differentiable, and its gradient is $L$–Lipschitz continuous (see Assumptions 4.1–4.2). We assume that we have $n$ workers available to work in parallel, each able to compute independent, unbiased stochastic gradients of $f$, whose variance is bounded by $\sigma^2$ (see Assumption 4.3). In this paper, we are interested in investigating the time complexity of methods working in this natural setup.

### 1.1 Parallel methods

With access to $n$ clients capable of computing stochastic gradients in parallel, perhaps the most naive and classical approach is running Minibatch SGD (Cotter et al., 2011; Goyal et al., 2017; Gower et al., 2019).

**Minibatch SGD.** This method awaits the completion of all workers' computations of a single stochastic gradient before executing a gradient-type step:

1. receive a single stochastic gradient $\nabla f(x^k; \xi_i^k)$ from each worker $i \in [n]$,
2. update the model via $x^{k+1} = x^k - \gamma \frac{1}{n} \sum_{i=1}^n \nabla f(x^k; \xi_i^k)$,

where $[n] := \{1, \ldots, n\}$, $\gamma > 0$ is a stepsize, $\xi_i^k$ are i.i.d. samples from $\mathcal{D}$, and the gradients $\nabla f(x^k; \xi_i^k)$ are calculated in parallel.

In real systems, each worker's computational power may differ from the others, leading to varying completion times of gradient computation. A notable drawback of Minibatch SGD is its failure to account for these differences in compute times across workers. The duration of each step is determined by the slowest worker's computation time. As a result, all other workers remain idle after completing their tasks, waiting for the slowest device to finish. Meanwhile, this idle time could potentially be used in a more efficient way to improve the overall time complexity. Clearly, a redesign of the algorithm is necessary.

**Asynchronous SGD.** As a result, a new generation of algorithms emerged, known as asynchronous stochastic gradient descent (ASGD) methods, designed to fully utilize all available computational resources (Recht et al., 2011; Feyzmahdavian et al., 2016; Nguyen et al., 2018; Arjevani et al., 2020; Cohen et al., 2021; Mishchenko et al., 2022; Koloskova et al., 2022; Islamov et al., 2023).

Here, the server performs a gradient-type update immediately after receiving a stochastic gradient from any worker, without waiting for the others. The updated model is then sent back to the worker, which immediately begins computing a new stochastic gradient based on the updated model. By the time the worker finishes computing this gradient, the model may have already been updated multiple times on the server due to gradients received from other workers. This creates a delay in the model update, denoted as $\delta_k$. The algorithm can be described as follows:

1. receive a stochastic gradient $\nabla f(x^{k-\delta_k}; \xi^{k-\delta_k})$ from any worker,

2. update the model via $x^{k+1} = x^k - \gamma \nabla f(x^{k-\delta_k}; \xi^{k-\delta_k})$,

3. send new $x^{k+1}$ to the worker so the worker computes $\nabla f(x^{k+1}; \xi^{k+1})$.

Cohen et al. (2021); Mishchenko et al. (2022); Koloskova et al. (2022) showed that ASGD is provably faster in terms of time complexity then Minibatch SGD.

However, it turns out that this untamed and wild asynchrony can be detrimental. The drawback of ASGD lies in the assumption that all workers' computations are beneficial. It suffers from the issue of updating the model with potentially significantly delayed gradients, which ultimately harms convergence and, consequently, the overall time complexity, as discussed in the work of Tyurin & Richtárik (2024). To address this issue, there was a need to introduce a method that ignores outdated gradients while maintaining the philosophy of maximizing the utilization of available computational resources.

**Rennala SGD.** Such a method was proposed in a recent breakthrough by Tyurin & Richtárik (2024). Their method which can be viewed as a modification of the Minibatch SGD method. At each iteration the server collects a batch of gradients, but it allows workers to send as many gradients as they can on the same point $x^k$. Then, using this batch, Rennala SGD proceeds with a gradient-type update using this batch as in Minibatch SGD:

1. wait until the server receives $B$ stochastic gradients at point $x^k$,

2. update the model via $x^{k+1} = x^k - \gamma \frac{1}{B} \sum_{j=1}^{B} \nabla f(x^k; \xi_j^k)$,

more details on Rennala SGD are in Appendix F. In this case, the faster the worker, the more gradients it sends. For the struggling workers, it may happen that they are completely ignored.

Their approach considers a setting where each worker $i$ requires a fixed $\tau_i > 0$ seconds to compute a stochastic gradient. For the first time lower bounds on time complexity were obtained for first order ASGD methods in the above mentioned fixed compute time regime for nonconvex functions with Lipschitz gradients. They showed that Rennala SGD is mini-max optimal in this setup in terms of time complexity.

While it may seem that the story is over, we want to question the fixed time assumption, arguing that a random time model is more realistic. The claim of optimality does not hold because of this randomness, suggesting that the algorithms need to be reevaluated and redesigned. We believe that a redesign is necessary to better fit this more realistic approach.

## 2 PROBLEM SETUP AND CONTRIBUTIONS

The deterministic compute time setup considered by Tyurin & Richtárik (2024), where Rennala SGD is optimal, fails to capture the complexities of real-world distributed learning environments. In practice, compute times are often uncertain due to various factors such as failing hardware, preemption by other jobs, delays in GPU computation, and inconsistencies in network communications (Chen et al., 2016; Dutta et al., 2018). This uncertainty is even more pronounced in federated learning scenarios, where client unreliability can lead to unpredictable computation times or even incomplete tasks (Kairouz et al., 2021).

To address these real-world challenges, we propose a more practical setup that incorporates randomness into compute times. Specifically, we consider a scenario where the stochastic gradient computation time of worker $i$ is given by:

$$\tau_i + \eta_i, \tag{2}$$

where $\tau_i > 0$ is a constant representing the minimum time for client $i$ to complete the gradient computation, and $\eta_i$ is a non-negative random variable drawn from some distribution $\mathcal{J}_i$, modeling the aforementioned uncertainties.

In this more realistic setting, existing methods like Rennala SGD and ASGD can perform poorly or even fail to converge. We can illustrate this with a simple example:

Consider a scenario where each time we request a device to compute a stochastic gradient, one of two outcomes occurs. Either the device completes the computation exactly after the minimum time $\tau$ without any delays, or something goes wrong and the computation is never completed. This situation can be modeled using a random time $\eta$ as follows:

$$\eta = \begin{cases} 0, & \text{with probability } 1 - q, \\ \infty^1, & \text{with probability } q, \end{cases} \tag{3}$$

where $0 < q < 1$. In this scenario, any method that waits for a certain number of batches on each iteration to perform a step runs the risk of never receiving the required batch and getting stuck. This includes methods like Rennala SGD or ASGD. Specifically, if the algorithm waits for a single stochastic gradient on each iteration, there is a probability $q^n$ that it will never receive it and consequently never proceed.

To address these limitations, we propose a new method that, unlike Rennala SGD or ASGD, does not wait for a fixed number of gradients (such as one in ASGD). Instead, it allocates a specific time for computing each stochastic gradient. If a client fails to complete its computation within the designated time, the partial computation is discarded, and a new computation is initiated. Our main contributions are as follows.

- In Section 4, we propose a new time efficient asynchronous parallel SGD method MindFlayer SGD Algorithm 1 for the heterogeneous and random worker compute times regime (Equation (2)). To the best of our knowledge, MindFlayer SGD is the first algorithm designed to work in this regime. We show that our method is a generalization of Rennala SGD, meaning that it is optimal in the deterministic compute times setup.

- In Section 5, we show that the theoretical time complexity of MindFlayer SGD can be arbitrarily faster than that of Rennala SGD or ASGD, depending on the distributions of computation times. Specifically, we demonstrate that if the distributions of computation times $\mathcal{J}_i$ are positively skewed, our method is faster, with the performance gap increasing as the skewness coefficient grows. As shown in Figure 1, where $\mathcal{J}_i = \text{Lognormal}(0, s)$. As $s$ gets bigger, the distribution's skewness coefficient gets bigger and the performance of Rennala SGD or ASGD gets worse. Meanwhile, our method MindFlayer SGD is robust to the change of the variance.

- In Section 6, we experimentally validate this performance. We provide practical guidelines for using MindFlayer SGD, and demonstrate its superiority over Rennala SGD and ASGD. We conduct evaluations using various functions and distributions. For distributions, we consider Lognormal, Log-Cauchy, and the Infinite-Bernoulli (defined by Equation (3))

---

[1]We can view $\eta$ as an extended real random variable, or just assume that $\infty$ is a very big number.

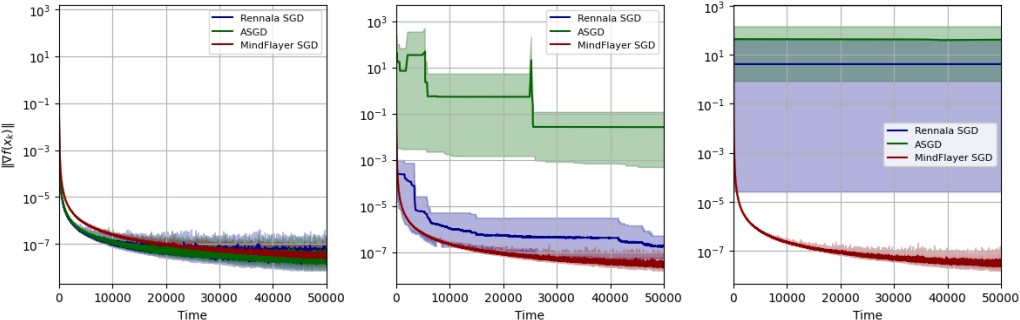

Figure 1: We ran an empirical experiment[2] where we employ the same $\mathcal{J}_i = \text{Lognormal}(0, s)$ distribution for all clients $i \in [n]$, with varying standard deviations $s$. Specifically, we set $s = 1$ for the left, $s = 10$ for the middle, and $s = 100$ for the right. Additionally, we set $\tau_i = \sqrt{i+1}$. As we observe, with an increase in the variance of the distribution, MindFlayer SGD demonstrates the ability to significantly outperform Rennala SGD and ASGD.

distributions. Regarding the functions, we consider a quadratic loss and a neural network on the MNIST (LeCun et al., 1998) dataset. This diverse testing setup enables us to showcase MindFlayer SGD's robustness and effectiveness across various challenging scenarios.

- In Appendix D, we expand our theory to develop Vecna SGD, designed for the heterogeneous case, where workers have datasets that are coming from different distributions.

- In Appendix E, we present a simple modification of our algorithm, Rennala SGD, which we call Mod MindFlayer SGD. This version is more suitable for practical implementation.

## 3 MOTIVATION AND SINGLE DEVICE CASE

To illustrate the motivation behind the design of our new method, let us consider a single device setup. Recall the scenario introduced in Equation (3) where we have single device and it either returns a gradient after $\tau$ time or gets stuck with probability $q$. A straightforward and optimal workaround to this issue is to wait exactly $\tau$ seconds. If we do not receive a gradient within this time frame, it indicates that we will never receive it, so there is no point in waiting longer. In this case, we discard the current computation, which would take forever anyway, and request the device to compute the gradient again. The probability of getting stuck again is lower, so eventually, we will receive a gradient and move forward.

More generally, consider the following two strategies for each step:

- **Strategy 1:** Rennala SGD. We wait for the first $B$ stochastic gradients. Thus, the time for one step for this strategy is the random variable:

$$T_B = \sum_{j=1}^{B} (\tau + \eta^j).$$

- **Strategy 2:** MindFlayer SGD. We repeat the following random trial $B$ times: allocate time $t$ for computing a stochastic gradient. If we do not receive a stochastic gradient within that time, discard the current computation and start over. Then the time for the $j$-th trial is given by:

$$T^j(t) = \begin{cases} \tau + \eta^j, & \text{if } \eta^j \leq t, \\ \tau + t, & \text{if } \eta^j > t. \end{cases}$$

Thus, the time for one step for this strategy is the random variable:

$$\tilde{T}_B(t) = \sum_{j=1}^{B} T^j(t).$$

---

[2]On a quadratic problem with $n = 5$ clients. We tuned stepsizes for all, and used theoretical trials $B_i$ for MindFlayer SGD from Theorem 4.5 and tuned batch size for Rennala SGD, see Section 6.

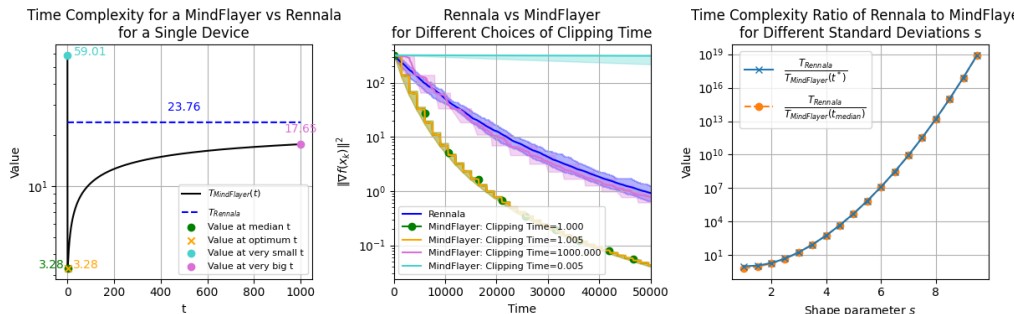

Figure 2: **On the left**, we compare the time complexity of MindFlayer SGD as a function of clipping time ($t$) against the constant time complexity of Rennala SGD, demonstrating the adaptive efficiency of MindFlayer SGD at various choices of $t$. **In the middle**, empirical validation[3] is shown where the reduction in time complexity for MindFlayer SGD is tested using the same clipping times as in the left graph, illustrating consistent performance improvements. **On the right**, the ratio of time complexities between Rennala SGD and MindFlayer SGD is plotted across different standard deviations ($s$), revealing exponential efficiency gains for MindFlayer SGD at optimal clipping times, with trends at median clipping times reflecting similar efficiencies.

In the second case, rather than waiting for $B$ gradients, we attempt to compute $B$ gradients. Essentially, we limit the time spent on computing a stochastic gradient. In expectation, Strategy 2 will collect $Bp$ gradients per iteration, where $p = P(\eta \leq t)$ is the probability of collecting a gradient within a trial. Setting $t = \infty$ removes this restriction, resulting in the same strategy as the first one, Rennala SGD.

For MindFlayer SGD, each iteration, on average, receives only $Bp$ gradients, making it effectively a scaled-down version of Rennala SGD. Consequently, MindFlayer SGD is expected to require $1/p$ times more iterations than Rennala SGD to achieve the same level of convergence. We have the following proposition.

**Proposition 3.1** (Proof in Appendix H). *Let $K$ be the number of iterations required by Rennala SGD to find an $\varepsilon$-stationary point. Then, for sufficiently small $\varepsilon$, MindFlayer SGD needs $K/p$ iterations to find an $\varepsilon$-stationary point.*

Thus, the time complexities in this setting are given by:

$$T_{\text{RennalaSGD}} = K\mathbb{E}\left[T_B\right] = KB(\tau + \mathbb{E}\left[\eta\right]),$$

$$T_{\text{MindFlayerSGD}}(t) = \frac{K}{p}\mathbb{E}\left[\tilde{T}_B(t)\right] = \frac{K}{p}B(\tau + (1-p)t + p\mathbb{E}\left[\tau|\tau \leq t\right]) \leq \frac{K}{p}B(\tau + t).$$

This leads us to the following remark.

*Remark* 3.2. For the case where $n = 1$, MindFlayer SGD is faster than Rennala SGD if there exists a time threshold $t > 0$ such that the following inequality holds:

$$\frac{\tau + t}{P(\eta \leq t)} < \tau + \mathbb{E}\left[\eta\right].$$

It is important to note that this can hold for a wide range of values of $t$, including any finite value. The latter is particularly relevant in cases where $\mathbb{E}\left[\eta\right] = \infty$. An example of such a scenario is illustrated in Equation (3). There are many other distributions for which the expectation is not finite, such as the Log-Cauchy distribution, Lévy distribution, Log-t distribution, Landau distribution, and so forth.

A less restrictive example of distributions are positively skewed distributions. Let $s = \mathbb{E}\left[\eta\right] - \text{Med}[\eta]$ be the skewness coefficient of the distribution $\mathcal{J}$. If $s > 0$ we say that the distribution is positively skewed. Then we have the following proposition.

**Proposition 3.3.** *[Proof in Appendix H] For the $n = 1$ case, if $s > \tau + \text{Med}[\eta]$ then MindFlayer SGD is faster than Rennala SGD. Moreover, if $s = (\tau + \text{Med}[\eta])(2\alpha - 1)$ then*

$$\frac{T_{\text{RennalaSGD}}}{T_{\text{MindFlayerSGD}}(\text{Med}[\eta])} \geq \alpha.$$

---

[3]On a quadratic problem with theoretical hyperparameters, see Section 6.

Therefore, Rennala SGD can be arbitrarily bad. As an example consider the Lognormal$(\mu, \sigma^2)$ distribution. For this distribution, we have:

$$s = \mathbb{E}\left[\eta\right] - \text{Med}[\eta] = \exp\left(\mu + \frac{\sigma^2}{2}\right) - \exp(\mu).$$

Thus, as we increase $\sigma$, the difference becomes arbitrarily large.

To verify this, we also conducted a small experiment, see Figure 2. The right plot showcases how the ratio of time complexity between Rennala SGD and MindFlayer SGD can get arbitrarily large for the optimal clipping time $t^* := \arg\min_t T_{\text{MindFlayerSGD}}(t)$ and even the median of the distribution $t_{\text{median}} = \text{Med}[\eta]$. The left and middle plots showcase the potential improvement, and even loss from choosing different clipping times $t$.

## 4 MINDFLAYER SGD

Here, we propose our MindFlayer SGD algorithm for multiple device case ($n > 1$). For the heterogeneous case, please refer to Appendix D.

---

**Algorithm 1** MindFlayer SGD [4]

1: **Input:** starting point $x^0 \in \mathbb{R}^d$, stepsize $\gamma > 0$, allotted times $t_1, \ldots, t_n \geq 0$, number of trials per client $B_1, \ldots, B_n \geq 0$
2: **for** $k = 1, 2, \ldots, K$ **do**
3:     Put $g^k = 0$
4:     Send $x^k$ to all clients
5:     Run Method 2 in all clients $i = 1, 2, \ldots, n$
6:     **while** there is a client that has trials to perform **do**
7:         Wait for the fastest client
8:         Receive gradient $g$
9:         $g^k = g^k + g$
10:     **end while**
11:     $g^k = \frac{g^k}{B}$, $\diamond$ $B = \sum_{i=1}^n p_i B_i$ and $p_i = F_i(t_i) = P(\eta_i \leq t_i)$.
12:     $x^{k+1} = x^k - \gamma g^k$
13: **end for**

---

**Algorithm 2** Client $i$-s $k$-th step

1: Receive $x^k$ from the server
2: **for** $j = 1, 2, \ldots, B_i$ **do**
3:     Sample $\eta_i^j \sim \mathcal{J}_i$
4:     **if** $\eta_i^j \leq t_i$ **then**
5:         $g = \nabla f(x^k; \xi_i^j), \; \xi_i^j \sim \mathcal{D}$
6:         Send $g$ to the server
7:     **end if**
8: **end for**

---

The MindFlayer SGD algorithm begins with an initialization at a starting point $x^0$ in $\mathbb{R}^d$, with a specified stepsize $\gamma > 0$, time allowances $t_i > 0$, and trial counts $B_i \geq 0$ for each client. In each iteration $k$, ranging from $k = 1$ to $K$, the server distributes the current point $x^k$ to all clients. Each client $i$ then executes a subroutine (Algorithm 2) to attempt to compute $B_i$ stochastic gradients from samples $\xi_i^j$ drawn from a distribution $\mathcal{D}$. During each attempt, client $i$ starts computing a stochastic gradient; if the computation exceeds the allotted time $t_i$, they discard the current gradient and begin another computation. Consequently, the actual number of stochastic gradients received from each client $i$ becomes a random variable, ranging from 0 to $B_i$. The expected number of gradients from client $i$ is given by $p_i B_i$, leading to an overall expected total of stochastic gradients $B = \sum_{i=1}^n p_i B_i$. The server aggregates these received stochastic gradients and normalizes the collective gradient by the expected batch size $B$. Finally, the point is updated to $x^{k+1} = x^k - \gamma g^k$ following each aggregation round.

In the special case where the computation time is deterministic, i.e., $\eta_i = 0$ for every worker $i \in [n]$, we have $p_i = 1$ for all $i$. While Rennala SGD does not explicitly specify the number of gradient computations $B_i$ for each client, in the deterministic setting, each client will send a fixed number of gradients per communication round. Consequently, for any $t > 0$, MindFlayer SGD Algorithm 1, by choosing $B_i$ appropriately, reduces to Rennala SGD Algorithm 7.

---

[4]We name our method MindFlayer SGD, drawing inspiration from The Mind Flayer from *Stranger Things*, due to its ability to precisely control its clients (Algorithm 2), analogous to the creature's supreme control over its victims (The Flayed).

However, the situation changes when $\eta_i > 0$ is not a constant random variable. If we set $t_i = \infty$ for all $i \in [n]$, MindFlayer SGD Algorithm 1 does not reduce to Rennala SGD Algorithm 7. This is because, in the case of Rennala SGD, the randomness in each iteration causes the number of stochastic gradients computed by each client to vary across different communication rounds. Nevertheless, this scenario is not our primary focus, as we will demonstrate that allowing each worker to complete its gradient computation by setting $t_i = \infty$ is inefficient when dealing with positively skewed distributions.

To continue with the analysis of MindFlayer SGD, we first present the assumptions under which this method is studied.

### 4.1 ASSUMPTIONS

We consider standard assumptions used in the nonconvex optimization.

**Assumption 4.1.** Function $f$ is differentiable, and its gradient is $L$–Lipschitz continuous, i.e., $\|\nabla f(x) - \nabla f(y)\| \leq L \|x - y\|$, for all $x, y \in \mathbb{R}^d$.

**Assumption 4.2.** There exist $f^{\mathrm{inf}} \in \mathbb{R}$ such that $f(x) \geq f^{\mathrm{inf}}$ for all $x \in \mathbb{R}^d$.

**Assumption 4.3.** For all $x \in \mathbb{R}^d$, stochastic gradients $\nabla f(x; \xi)$ are unbiased and $\sigma^2$-variance-bounded, i.e., $\mathbb{E}_\xi [\nabla f(x; \xi)] = \nabla f(x)$ and $\mathbb{E}_\xi \left[ \|\nabla f(x; \xi) - \nabla f(x)\|^2 \right] \leq \sigma^2$, where $\sigma^2 \geq 0$.

### 4.2 CONVERGENCE THEORY

The following theorem gives iterations guarantees for the convergence of MindFlayer SGD.

Even though MindFlayer SGD is similar to Rennala SGD the convergence analysis require additional considerations, since the batch size is a random variable here as apposed to the case of Rennala SGD.

**Theorem 4.4.** *Assume that Assumptions 4.1, 4.2 and 4.3 hold. Let $B = \sum_{i=1}^n p_i B_i$ and $\gamma = \frac{1}{2L} \min \left\{ 1, \frac{\varepsilon B}{\sigma^2} \right\}$ in Algorithm 1. Then, after*

$$K \geq \max \left\{ 1, \frac{\sigma^2}{\varepsilon B} \right\} \frac{8L \left( f(x^0) - f^{\mathrm{inf}} \right)}{\varepsilon}$$

*iterations, the method guarantees that $\frac{1}{K} \sum_{k=0}^{K-1} \mathbb{E} \left[ \|\nabla f(x^k)\|^2 \right] \leq \varepsilon$.*

*Sketch of Proof.* (Complete proof in Appendix I.1) We consider Algorithm 1 as a conventional SGD using the following gradient estimator:

$$g(x) = \frac{1}{B} \sum_{i=1}^n \sum_{j=1}^{B_i} I(\eta_i^j \leq t_i) \nabla f(x; \xi_i^j),$$

where $I(\cdot)$ denotes the indicator function. Prior to applying the classical SGD theorem (Theorem G.2), it is essential to verify that this estimator meets the theorem's conditions, namely unbiasedness and a specific bound on the second moment of $g(x)$. We demonstrate that the estimator is unbiased, and that

$$\mathbb{E} \left[ \|g(x)^2\| \right] \leq 2 \|\nabla f(x)\|^2 + \frac{1}{B} \sigma^2.$$

With these conditions satisfied, we can proceed to apply Theorem G.2. $\qquad \square$

Note that in the deterministic case where $\eta_i = 0$ for all $i \in [n]$, we have $p_i = P(\eta_i \leq t_i) = 1$ for all $i \in [n]$. Therefore, we derive

$$K \geq \max \left\{ 1, \frac{\sigma^2}{\varepsilon B} \right\} \frac{8L \left( f(x^0) - f^{\mathrm{inf}} \right)}{\varepsilon},$$

with $B = \sum_{i=1}^n B_i$, yielding the same result as Rennala SGD, up to a constant factor.

We also achieve the same rate as $t_i \to \infty$ for all $i$, since in that scenario $p_i \to 1$. This is expected because we will observe a consistent number of stochastic gradients each time, though the timing may vary, as mentioned earlier.

However, if $t_i = 0$ for all $i \in [n]$, then $K = \infty$. This result is anticipated since, in this case, the success probability is zero for all clients, and thus the server never receives stochastic gradients.

### 4.3 TIME COMPLEXITY

The following theorem gives time complexity for MindFlayer SGD.

**Theorem 4.5** (Proof in Appendix I.2). *Assume that Assumptions 4.1, 4.2 and 4.3 hold. Let $B = \sum_{i=1}^{n} p_i B_i$ and $\gamma = \frac{1}{2L} \min\left\{1, \frac{\varepsilon B}{\sigma^2}\right\}$ in Method 1. Let $t = (t_1, \ldots, t_n)$, $t_1, \ldots, t_n \geq 0$. Without loss of generality assume that $0 < \tau_1 + t_1 \leq \cdots \leq \tau_n + t_n$. Let*

$$t(m) = \left(\sum_{j=1}^{m} \frac{p_j}{\tau_j + t_j}\right)^{-1} \left(S + \sum_{j=1}^{m} p_j\right),$$

*where $S = \max\left\{1, \frac{\sigma^2}{\varepsilon}\right\}$. Let $m^* = \arg\min_{m \in [n]} t(m)$, if there are several minimizers we take the smallest one. Put*

$$B_i = \lceil b_i \rceil, \quad b_i = \begin{cases} \frac{t(m^*)}{\tau_i + t_i} - 1, & \text{if } i \leq m^*, \\ 0, & \text{if } i > m^*. \end{cases}$$

*Then, MindFlayer SGD guarantees to find an $\epsilon$-stationary point after*

$$T_{\text{MindFlayerSGD}}(t) \geq 8 \times \min_{m \in [n]} \left\{ \left(\frac{1}{m} \sum_{j=1}^{m} \frac{p_j}{\tau_j + t_j}\right)^{-1} \left(\frac{S}{m} + \frac{1}{m} \sum_{j=1}^{m} p_j\right) \frac{\Delta L}{\varepsilon} \right\}$$

*seconds, where $\Delta = f(x_0) - f^{\inf}$.*

The theorem indicates that the optimal strategy is to disregard devices with a high value of $\tau_i + t_i / p_i$. Therefore, we should prioritize devices that not only have a high probability $p_i$ of completing the gradient within the allotted time $t_i$ but also have a relatively small sum of $\tau_i + t_i$. This approach is logical as it avoids including devices with substantial computation times and low probabilities of completing their tasks within the specified duration.

In the deterministic case where $\eta_i = 0$ for all $i \in [n]$, we have $p_i = 1$ for all $i$. Consequently, the time complexity of MindFlayer SGD at time $t$ is given by

$$T_{\text{MindFlayerSGD}}(t) \geq 8 \times \min_{m \in [n]} \left\{ \left(\frac{1}{m} \sum_{j=1}^{m} \frac{1}{\tau_j + t_j}\right)^{-1} \left(\frac{S}{m} + 1\right) \frac{\Delta L}{\varepsilon} \right\}.$$

Thus, the optimal choice of $t_i$ is $t_i = 0$ for all $i \in [n]$. Therefore, the final time complexity becomes

$$T_{\text{MindFlayerSGD}}(t) \geq 8 \times \min_{m \in [n]} \left\{ \left(\frac{1}{m} \sum_{j=1}^{m} \frac{1}{\tau_j}\right)^{-1} \left(\frac{1}{m} + 1\right) \frac{\Delta L}{\varepsilon} \right\}.$$

This formulation recovers the time complexity for Rennala SGD.

We still have the freedom to choose the $t_i$ allocation times. The optimal strategy would be to select them in a manner that minimizes the time complexity. As observed in Figure 2, setting $t_i = \text{Med}\,[\eta_i]$ proves to be a viable choice. This is further confirmed by our experiments in Section 6.

## 5 COMPARING TO RENNALA SGD

Comparing the theoretical performance of Rennala SGD and MindFlayer SGD is particularly challenging due to the inherent randomness in the time complexity of Rennala SGD and the dependence of MindFlayer SGD on optimizing time variables $t_i$. For example, a comparison using the expected time complexity may fail to capture the nuances of each algorithm's performance across different distributions. Thus, we turn to an empirical comparison to provide insights into their practical behavior. In particular, we aim to demonstrate how MindFlayer SGD can achieve arbitrarily better performance in scenarios where the distributions exhibit high variance or heavy tails (see Figure 3).

To begin, we derive the time complexity of Rennala SGD in the context of random times. Let $\mathcal{B} := \{(B_1, B_2, \ldots, B_n) : B_i \in \mathbb{N}_0; \sum_{i=1}^{n} B_i = B\}$ be the set of all possible batch sizes for each device, the time $T_B$ required for one step with batch size $B$ of Rennala SGD is given by:

$$T_B = \min_{\mathcal{B}} \left\{ \max_{i \in [1,n]} \left\{ B_i \tau_i + \sum_{j=1}^{B_i} \eta_i^j \right\} \right\} \geq T_1 \tag{4}$$

$$= \min_{i \in [n]} \left\{ \tau_i + \eta_i^1 \right\} \geq \min_{i \in [n]} \left\{ \tau_i \right\} + \min_{i \in [n]} \left\{ \eta_i^1 \right\}.$$

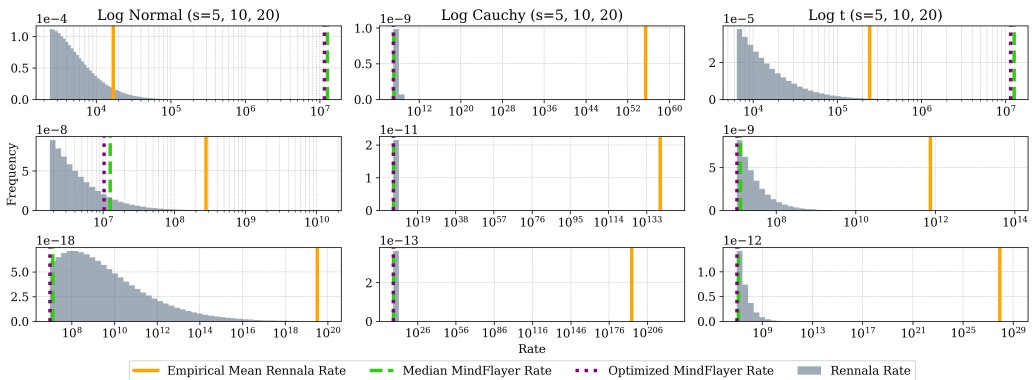

Figure 3: Empirical comparison of the performance rates between Rennala SGD and MindFlayer SGD is illustrated, as described in the corresponding sect on. We investigate three distributions: lognormal, log Cauchy, and log $t$ with 5 degrees of freedom. As the variance increases, the theoretical rate of MindFlayer SGD significantly outperforms that of Rennala SGD.

Thus, the expected time to collect a batch $B$ is

$$\mathbb{E}\left[T_B\right] \geq \tau_{\min} + \mathbb{E}\left[\min_{i \in [n]} \eta_i\right],$$

Note that if the distribution of $\min_{i \in [n]} \eta_i$ is heavy-tailed, then the expected time complexity becomes infinite, thus favoring MindFlayer SGD over Rennala SGD. A simple illustration of this occurs when extending the Equation (3) case, where $\eta$ is either zero or infinite, to scenarios involving multiple devices. In such cases, the expectation of the minimum time across devices, $\min_{i \in [n]} \eta_i$, also results in an infinite expected time complexity.

While a detailed theoretical comparison is intractable, we conduct an empirical comparison to highlight practical differences between the two algorithms. To capture the randomness of Rennala SGD's rate, we generate a histogram: we create a histogram for $T_B$ and then convolving it $K$ times with itself. Where $K$ is the number of iterations required for $\epsilon$-convergence.

The time complexity of Rennala SGD is a random variable that is the sum of $K$ copies of $T_B$, where is $K$ is number of iterations to get $\epsilon$-convergence.

For MindFlayer SGD, we evaluate two strategies for selecting $t_i$: (1) using the median of the distributions $\mathcal{J}_i$, and (2) solving the following optimization problem:

Fix $m \in [n]$, minimize $t(m)$ over $t = (t_1, \cdots, t_n)$, (remember $p_j = F_j(t_j)$).

We optimize this using the L-BFGS-B algorithm, a well-suited method for solving smooth, convex, or mildly nonconvex problems due to its efficiency and robustness (Zhu et al., 1997). For each $m$, we take the minimum over all possible configurations.

Our empirical results, illustrated in Figure 3, demonstrate that as the variance of the underlying distribution increases, MindFlayer SGD consistently outperforms Rennala SGD. The heavy-tailed nature of the distributions causes Rennala SGD to experience extreme slowdowns, while MindFlayer SGD maintains robust performance.

## 6 EXPERIMENTS

In this section we explain the setup for comparing MindFlayer SGD, Rennala SGD, and ASGD, which we used throughout this paper. We compare the algorithms' performance on a quadratic optimization (5) task with access to a stochastic gradient. The parallelism was simulated on a machine with 2 Intel(R) Xeon(R) Gold 6226R CPUs @ 2.90GHz, with a total of 64 logical CPUs. For each setting of the algorithm, we run 10 different seeds for the random time and plot the average, minimum and maximum, see Figure 1, Figure 2, etc.

We use a similar setup to the one employed by Tyurin & Richtárik (2024), but modify it so that we have a known expected variance. We make this choice, so we can compare theoretical parameters, as we did in Figure 2.

Furthermore, we consider the homogeneous optimization problem 1, with the convex quadratic function:

$$f(x) = \frac{1}{2}x^\top A x - b^\top x \qquad \forall x \in \mathbb{R}^d.$$

We take $d = 1000$,

$$A = \frac{1}{4}\begin{bmatrix} 2 & -1 & & 0 \\ -1 & \ddots & \ddots & \\ & \ddots & \ddots & -1 \\ 0 & & -1 & 2 \end{bmatrix} \in \mathbb{R}^{d \times d} \quad \text{and} \quad b = \frac{1}{4}\begin{bmatrix} -1 \\ 0 \\ \vdots \\ 0 \end{bmatrix} \in \mathbb{R}^d. \tag{5}$$

Assume that all $n$ workers has access to the following unbiased stochastic gradients:

$$[\nabla f(x, \xi)]_j := \nabla_j f(x) + \xi,$$

where $\xi \sim \mathcal{N}(0, 0.0003^2)$, thus, we get that in Assumption 4.3 we have,

$$\sigma^2 = 0.0003^2 \cdot d = 0.0003^2 \cdot 1000.$$

Now setting the convergence threshold $\epsilon = 10^{-4}$, we can infer all theoretical parameters. To find the optimal time corresponding to Rennala SGD we need to fix the times, we do that by either removing the randomness, or adding the expected randomness. On the other hand, for MindFlayer SGD we use the results from Theorem 4.5 to set the theoretical number of trials for each client. For some experiments we used theoritical stepsizes, e.g. Figure 2, for others we used the range of stepsizes from a set $\{2^i | i \in [-10, 10]\}$, e.g. Figures 1, 5, and 5, similarly to Tyurin & Richtárik (2024). Finally, for the nonconvex problem in Figure 6 we tried the set $\{0.01, 0.001, 0.0001\}$.

In addition to the experimental results shown throughout the paper, we ran two more experiments. One with the Infinite-Bernoulli distribution on the same quadratic problem, and a second with the Log-Cauchy distribution with a small two-layer neural network on the MNSIT dataset, see Figure 5 and Figure 6.

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

CONTENTS

## A    RELATED WORK

There are several other related works. Dutta et al. (2018) explore the error-runtime trade-offs in distributed SGD, revealing how slower and stale gradients can sometimes enhance convergence processes. Woodworth et al. (2020) compare local SGD with minibatch SGD, analyzing the efficiency of local updates in different distributed settings. Wu et al. (2022) advance the understanding of asynchronous methods by proposing delay-adaptive step-sizes that adjust to asynchronous learning environments, optimizing the convergence rates. Furthermore, Hanna et al. (2022; 2020) focus on adaptive stochastic gradient descent to improve communication efficiency in distributed learning, offering strategies that reduce communication demands while maintaining fast convergence.

## B    CONCLUSION AND FUTURE WORK

In this paper, we address the problem of minimizing the expectation of nonconvex functions with Lipschitz gradients, with the use of parallel workers computing stochastic gradients. Our focus lies on the challenging scenario where worker compute times are heterogeneous and random, expanding on recent developments in ASGD methods like Rennala SGD. We observe that while Rennala SGD performs optimally in environments with deterministic compute times, its effectiveness diminishes under random compute conditions.

To better understand and improve stochastic optimization in these conditions, we introduce a novel asynchronous SGD method named MindFlayer SGD. This method adjusts to the randomness in computation times by not adhering to a fixed batch size but rather setting specific times for computing single stochastic gradients. If a client fails to deliver within this time frame, the computation is discarded, and the process restarts. This flexibility allows MindFlayer SGD to perform robustly across various conditions, notably outperforming both Rennala SGD and standard Asynchronous SGD (ASGD) in our theoretical and empirical analysis.

Our results demonstrate that MindFlayer SGD significantly reduces time complexity, particularly in environments characterized by positively skewed distribution of computation times. We empirically validate this in simulations with several distributions conditions where MindFlayer SGD consistently outperforms the other methods, particularly in high-variance scenarios. This showcases its superiority in adapting to the unpredictable duration of gradient computations typical in real-world applications such as federated learning environments.

In this study, our analysis was confined to computation times, with no consideration given to communication times. Future research will extend our investigation to include communication times. Moreover, we plan to explore the application of gradient estimators with varying variance bounds across different clients. We hypothesize that controlling these variance bounds could yield further benefits in the optimization process.

## C    TABLE OF NOTATIONS

| Notation | Meaning |
|---|---|
| $[n]$ | $\{1, \ldots, n\}$ |
| $L$ | Lipschitz constant of gradients, i.e., $\|\nabla f(x) - \nabla f(y)\| \leq L \|x - y\|$ (Assumption 4.1) |
| $f^{\mathrm{inf}}$ | Minimum value of the function, i.e., $f^{\mathrm{inf}} \leq f(x)$ (Assumption 4.2) |
| $\sigma^2$ | Variance bound on gradients, i.e., $\mathbb{E}_\xi \left[ \|\nabla f(x; \xi) - \nabla f(x)\|^2 \right] \leq \sigma^2$ (Assumption 4.3) |
| $\gamma$ | Stepsize |
| $\tau_i$ | Minimum time required for client $i$ to compute a gradient |
| $\eta_i$ | Additional random time taken while computing the gradient |
| $\mathcal{J}_i$ | Distribution of the non-negative random variable $\eta_i$ |
| $t_i$ | Allotted time for worker $i$ to compute a gradient |

# D HETEROGENEOUS REGIME

So far, we have discussed the scenario where all workers compute i.i.d. stochastic gradients. However, in distributed optimization and federated learning (Konečný et al., 2016), workers may have different datasets. Consider the following optimization problem:

$$\min_{x \in \mathbb{R}^d} \left\{ f(x) := \frac{1}{n} \sum_{i=1}^n \mathbb{E}_{\xi_i \sim \mathcal{D}_i} \left[ f_i(x; \xi_i) \right] \right\}, \tag{6}$$

where $f_i : \mathbb{R}^d \times \mathbb{S}_i \to \mathbb{R}^d$ and $\xi_i$ are random variables with some distributions $\mathcal{D}_i$ on $\mathbb{S}_i$. Problem (6) generalizes problem (1).

## D.1 RELATED WORK AND DISCUSSION

The optimization problem (6) has been thoroughly studied in many papers, including (Aytekin et al., 2016; Mishchenko et al., 2018; Nguyen et al., 2022; Wu et al., 2022; Koloskova et al., 2022; Mishchenko et al., 2022). There have been attempts to analyze Asynchronous SGD in the heterogeneous setting. For example, Mishchenko et al. (2022) demonstrated convergence only to a neighborhood of the solution. In general, achieving good rates for Asynchronous SGD is difficult without making additional assumptions about the similarity of the functions $f_i$ (Koloskova et al., 2022; Mishchenko et al., 2022).

In the deterministic case, when $\sigma^2 = 0$, Wu et al. (2022) analyzed the PIAG method in the deterministic heterogeneous regime and showed convergence. Although the performance of PIAG can be good in practice, in the worst case PIAG requires $O\left(\tau_n \widehat{L} \Delta / \varepsilon\right)$ seconds to converge, where $\tau_n$ is the time delay of the slowest worker, $\widehat{L} := \sqrt{\sum_{i=1}^n L_i^2}$, and $L_i$ is a Lipschitz constant of $\nabla f_i$. Note that the synchronous Minibatch SGD (see Section 1.1) method has the complexity $O\left(\tau_n L \Delta / \varepsilon\right)$, which is always better.[5]

Tyurin & Richtárik (2024) proposed an optimal method in the regime where worker computation times are deterministic, similar to the homogeneous setup.

## D.2 VECNA SGD

Here we describe our method called Vecna SGD.

---

**Algorithm 3** Vecna SGD [6]

---

1: **Input:** starting point $x^0 \in \mathbb{R}^d$, stepsize $\gamma > 0$, allotted times $t_1, \ldots, t_n \geq 0$, number of trials per client $B_1, \ldots, B_n \geq 0$
2: **for** $k = 1, 2, \ldots, K$ **do**
3:     Put $g_i^k = 0$
4:     Send $x^k$ to all clients
5:     Run Method 4 in all clients $i = 1, 2, \ldots, n$
6:     **while** there is a client that has trials to perform **do**
7:         Wait for the fastest client
8:         Receive gradient $g_i$ from client $i$
9:         $g_i^k = g_i^k + g$
10:     **end while**
11:     $g^k = \frac{1}{n} \sum_{i=1}^n \frac{g_i^k}{p_i B_i},$                $\diamond \ p_i = F_i(t_i) = P(\eta_i \leq t_i).$
12:     $x^{k+1} = x^k - \gamma g^k$
13: **end for**

---

---
[5]In the nonconvex case, $\widehat{L}$ can be arbitrarily larger than $L$.

---

**Algorithm 4** Client $i$-s $k$-th step

---

1: Receive $x^k$ from the server
2: **for** $j = 1, 2, \ldots, B_i$ **do**
3:     Sample $\eta_i^j \sim \mathcal{J}_i$                 $\diamond$ Start computing gradient estimator.
4:     **if** $\eta_i^j \leq t_i$ **then**
5:         $g = \nabla f(x^k; \xi_i^j), \ \xi_i^j \sim \mathcal{D}$    $\diamond$ The computation completes within the allotted time $t_i$.
6:         Send $g$ to the server
7:     **end if**
8: **end for**

---

The Vecna SGD algorithm begins with an initialization at a starting point $x^0$ in $\mathbb{R}^d$, with a specified stepsize $\gamma$, time allowances $t_i$, and trial counts $B_i$ for each client. In each iteration $k$, ranging from $k = 1$ to $K$, the server distributes the current point $x^k$ to all clients. Each client $i$ then executes a subroutine (Algorithm 4) to attempt to compute $B_i$ stochastic gradients from samples $\xi_i^j$ drawn from a distribution $\mathcal{D}$. During each attempt, client $i$ starts computing a stochastic gradient; if the computation exceeds the allotted time $t_i$, they discard the current gradient and begin another computation. Consequently, the actual number of stochastic gradients received from each client $i$ becomes a random variable, ranging from 0 to $B_i$. The expected number of gradients from client $i$ is given by $p_i B_i$. The server normalizes the gradients by the expected batch size $p_i B_i$ and then aggregates them. Finally, the point is updated to $x^{k+1} = x^k - \gamma g^k$ following each aggregation round.

## D.3   CONVERGENCE THEORY

The following theorem gives iterations guarantees for the convergence of Vecna SGD.

**Theorem D.1** (Proof in Appendix J.1). *Assume that Assumptions 4.1, 4.2 hold for the function $f$ and Assumption 4.3 holds for the function $f_i$ for all $i \in [n]$. Let $\gamma = \min\left\{\frac{1}{\sqrt{L\alpha K}}, \frac{1}{L\beta}, \frac{\varepsilon}{2L\zeta}\right\}$ in Algorithm 3. Then after*

$$K \geq \frac{12\Delta L}{\varepsilon} \max\left\{\beta, \frac{12\Delta\alpha}{\varepsilon}, \frac{2\zeta}{\varepsilon}\right\},$$

*iterations, the method guarantees that $\min_{0 \leq k \leq K} \mathbb{E}\left[\left\|\nabla f(x^k)\right\|^2\right] \leq \varepsilon$, where $\Delta = f(x_0) - f^{\inf}$ and*

$$\alpha = \frac{L}{n^2} \sum_{i=1}^{n} \frac{1 - p_i}{p_i B_i}, \quad \beta = 1, \quad \zeta = \frac{\sigma^2}{n^2} \sum_{i=1}^{n} \frac{1}{p_i B_i}.$$

## D.4   TIME COMPLEXITY

The following theorem gives time complexity for Vecna SGD.

**Theorem D.2** (Proof in Appendix J.2). *Assume that Assumptions 4.1, 4.2 hold for the function $f$ and Assumption 4.3 holds for the function $f_i$ for all $i \in [n]$. Let $\gamma = \min\left\{\frac{1}{\sqrt{L\alpha K}}, \frac{1}{L}, \frac{\varepsilon}{2L}\right\}$ in Algorithm 3, where*

$$\alpha = \frac{L}{n^2} \sum_{i=1}^{n} \frac{1 - p_i}{p_i B_i}, \quad \zeta = \frac{\sigma^2}{n^2} \sum_{i=1}^{n} \frac{1}{p_i B_i}.$$

*Let $t = (t_1, \ldots, t_n)$, $t_1, \ldots, t_n \geq 0$. Without loss of generality assume that $0 < \tau_1 + t_1 \leq \cdots \leq \tau_n + t_n$. Let*

$$T = \tau_n + t_n + \left[\frac{1}{n} \sum_{i=1}^{n} \frac{\tau_i + t_i}{p_i}\right] \frac{\sigma^2}{n\varepsilon} + \left[\frac{1}{n} \sum_{i=1}^{n} \frac{1 - p_i}{p_i}(\tau_i + t_i)\right] \frac{\Delta L}{n\varepsilon},$$

*where $\Delta = f(x_0) - f^{\inf}$. Put*

$$B_i = \lceil b_i \rceil, \quad b_i = \frac{T}{\tau_i + t_i}.$$

---

[6]We name our method Vecna SGD, drawing inspiration from Vecna from *Stranger Things*.

*Then,* Vecna SGD *guarantees to find an $\epsilon$-stationary point after*

$$T_{\text{VecnaSGD}}(t) \geq 288 \times \frac{\Delta L}{\varepsilon} \left( \tau_n + t_n + \left[ \frac{1}{n} \sum_{i=1}^{n} \frac{\tau_i + t_i}{p_i} \right] \frac{\sigma^2}{n\varepsilon} + \left[ \frac{1}{n} \sum_{i=1}^{n} \frac{1 - p_i}{p_i} \left( \tau_i + t_i \right) \right] \frac{\Delta L}{n\varepsilon} \right)$$

*seconds.*

# E  SIMPLIFYING MINDFLAYER FOR PRACTICAL USE

The version of MindFlayer SGD presented in this paper aims to be as general as possible, with the primary objective of providing theoretical insight, which is the focus of this work. Allowing for significant variability in the distributions of worker compute times intuitively necessitates the introduction of multiple hyperparameters, such as $B_i$ (batch sizes) and $t_i$ (clipping times), to ensure effective optimization under diverse scenarios. While these hyperparameters enable the algorithm to adapt to heterogeneous and random conditions, they also introduce additional complexity, which may complicate implementation in practical settings.

We propose Mod MindFlayer SGD, a practical variant that replaces $B_i$ and $t_i$ with two global parameters: a probabilistic threshold $p$, which reflects the likelihood of completing a gradient computation, and a global batch size $B$, specifying the total number of trials across all workers. This reformulation simplifies hyperparameter tuning while retaining robustness.

The parameter $p$ captures system reliability. For reliable systems, $p$ approaches 1, recovering Rennala SGD, while for less reliable systems, lower $p$ values leverage MindFlayer SGD 's robustness. The choice of $t_i$ can be guided by historical data via the inverse cumulative distribution function of $p$, or adjusted dynamically using the Robbins-Monro stochastic approximation, as such:

We update the clipping time $t_i$ at each iteration using the Robbins-Monro stochastic approximation (Robbins & Monro, 1951):

$$t_{i+1} = t_i - \alpha_i \left( I(T_i \leq t_i) - p \right)$$

where:

- $T_i$ is the observed compute time for the $i$-th iteration.

- $I(\cdot)$ is the indicator function, which is 1 if we don't clip, and 0 otherwise.

- $\alpha_i$ is a diminishing step size sequence, such as $\alpha_i = \frac{a}{i}$ with $a > 0$.

- $p$ is the target probability threshold.

Note that we do not need to know the exact value of $T_i$; we only require $I(T_i \leq t_i)$, which is 1 if the worker finishes the computation within the threshold and 0 otherwise.

By employing this dynamic adjustment, Mod MindFlayer SGD continuously adapts $t_i$ based on real-time observations of worker compute times, aligning the clipping threshold with the desired completion probability $p$. This method reduces the need for manual tuning of hyperparameters and enhances the algorithm's robustness to variability in compute times.

In Figure 4, we demonstrate that Mod MindFlayer SGD achieves comparable performance to MindFlayer SGD while simplifying hyperparameter selection, highlighting its practicality for distributed systems with heterogeneous and random worker compute times.

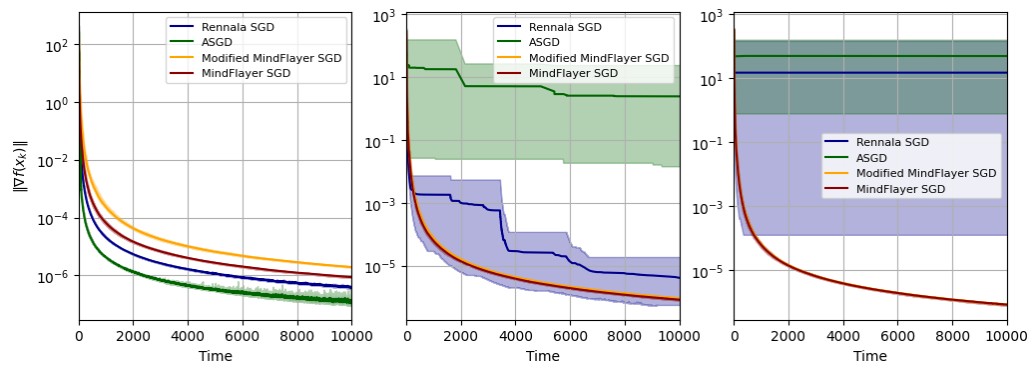

Figure 4: Here we recreate the setup from Figure 1, but add a hyperparameter tuned version of the Mod Mindflayer SGD.

---

**Algorithm 5** Mod MindFlayer SGD

1: **Input:** starting point $x^0 \in \mathbb{R}^d$, stepsize $\gamma > 0$, $p \in (0, 1]$, number of trails $B \geq 0$
2: **for** $k = 1, 2, \ldots, K$ **do**
3:      Put $g^k = 0$
4:      Send $x^k$ to all clients
5:      Run Method 6 in all clients $i = 1, 2, \ldots, n$ and stop old computations
6:      $b = 0$
7:      **while** $b < B$ **do**
8:          Wait for the fastest client
9:          Receive gradient $g$
10:        $g^k = g^k + g$
11:        $b = b + 1$
12:      **end while**
13:      $g^k = \frac{g^k}{B}$,
14:      $x^{k+1} = x^k - \gamma g^k$
15: **end for**

---

**Algorithm 6** Client $i$-s $k$-th step

1: Receive $x^k$ from the server
2: **while** True **do**
3:      Sample $\eta_i^j \sim \mathcal{J}_i$
4:      **if** $\tau_i + \eta_i^j \leq t_i$ **then**
5:          $g = \widehat{\nabla} f(x^k; \xi_i^j), \ \xi_i^j \sim \mathcal{D}$
6:          Send $g$ to the server
7:      **end if**
8: **end while**

## F THE RENNALA ALGORITHM

---

**Algorithm 7** Rennala SGD

1: **Input:** starting point $x^0$, stepsize $\gamma$, batch size $S$
2: Run Method 8 in all workers
3: **for** $k = 0, 1, \ldots, K - 1$ **do**
4:      Init $g^k = 0$ and $s = 1$
5:      **while** $s \leq S$ **do**
6:          Wait for the next worker
7:          Receive gradient and iteration index $(g, k')$
8:          **if** $k' = k$ **then**
9:             $g^k = g^k + \frac{1}{S} g; \quad s = s + 1$
10:        **end if**
11:        Send $(x^k, k)$ to the worker
12:      **end while**
13:      $x^{k+1} = x^k - \gamma g^k$
14: **end for**

---

**Algorithm 8** Worker's Infinite Loop

1: Init $g = 0$ and $k' = -1$
2: **while** True **do**
3:      Send $(g, k')$ to the server
4:      Receive $(x^k, k)$ from the server
5:      $k' = k$
6:      $g = \nabla f(x^k; \xi), \quad \xi \sim \mathcal{D}$
7: **end while**

We mention the Rennala SGD throughout the paper, here we provide a brief introduction to the method and its development. Algorithm 7 shows the work done by the server. Essentially, the server asynchronously waits to collect a batch of size $S$, whenever it receives a gradient from a worker that has the same iteration as the algorithm, it assigns it to compute a gradient at the same point $x_k$. After collecting the batch, we preform a synchronous update (given that all gradients were made on the same point $x_k$), using an average of the collected batch.

## G    THE CLASSICAL SGD THEORY

In this section, we present the classical SGD theory as developed by Ghadimi & Lan (2013) and Khaled & Richtárik (2020). Our analysis will follow the approach of the latter.

We consider the stochastic gradient descent (SGD) method:

$$x^{k+1} = x^k - \gamma g(x^k),$$

where $x^0 \in \mathbb{R}^d$ is the initial point, and $g(x)$ is a stochastic gradient estimator at $x$.

We make the following assumption:

**Assumption G.1.** The stochastic gradient estimator $g(x)$ satisfies:

$$\mathbb{E}\left[g(x)\right] = \nabla f(x)$$

$$\mathbb{E}\left[\|g(x)\|^2\right] \leq 2\alpha \left(f(x) - f^{\mathrm{inf}}\right) + \beta \|\nabla f(x)\|^2 + \zeta,$$

for all $x \in \mathbb{R}^d$ and some constants $\alpha, \beta, \zeta \geq 0$.

This assumption is both general and reasonable, and it is satisfied by many modern SGD-type methods. For further details, refer to Khaled & Richtárik (2020).

Under this assumption, we can derive the following convergence result.

**Theorem G.2** (Corollary 1 (Khaled & Richtárik, 2020))**.** *Assume that Assumptions 4.1, 4.2 and G.1 hold. Then for any $\varepsilon > 0$*

$$\min_{0 \leq k \leq K} \mathbb{E}\left[\|\nabla f(x^k)\|^2\right] \leq \varepsilon$$

*for*

$$\gamma = \min\left\{\frac{1}{\sqrt{L\alpha K}}, \frac{1}{L\beta}, \frac{\varepsilon}{2L\zeta}\right\},$$

*and*

$$K \geq \frac{12L\left(f(x_0) - f^{\mathrm{inf}}\right)}{\varepsilon} \max\left\{\beta, \frac{12\Delta\alpha}{\varepsilon}, \frac{2\zeta}{\varepsilon}\right\}.$$

## H    PROOFS FOR PROPOSITIONS IN SECTION 3

**Proposition 3.1.** *Let $K$ be the number of iterations required by Rennala SGD to find an $\varepsilon$-stationary point. Then, for sufficiently small $\varepsilon$, MindFlayer SGD needs $K/p$ iterations to find an $\varepsilon$-stationary point.*

*Proof.* The iterations of Rennala SGD can be viewed as iterations of Minibatch SGD. Thus, we can apply the classical SGD theory (Theorem G.2) to derive its iteration complexity:

$$K = \max\left\{1, \frac{\sigma^2}{\varepsilon B}\right\} \frac{8L(f(x^0) - f^{\mathrm{inf}})}{\varepsilon}.$$

For MindFlayer SGD, the iteration complexity follows from Theorem 4.4. Therefore, the number of iterations $K_M$ required for MindFlayer SGD to guarantee that

$$\frac{1}{K_M} \sum_{k=0}^{K_M - 1} \mathbb{E}\left[\|\nabla f(x^k)\|^2\right] \leq \varepsilon$$

is given by

$$K_M = \max\left\{1, \frac{\sigma^2}{\varepsilon B p}\right\} \frac{8L(f(x^0) - f^{\text{inf}})}{\varepsilon}.$$

If $\varepsilon \leq \frac{\sigma^2}{B}$, we have

$$K_M = \frac{K}{p}.$$

$\square$

**Proposition 3.3.** *For the $n = 1$ case, if $s > \tau + \text{Med}[\eta]$ then* MindFlayer SGD *is faster than* Rennala SGD. *Moreover, if $s = (\tau + \text{Med}[\eta])(2\alpha - 1)$ then*

$$\frac{T_{\text{RennalaSGD}}}{T_{\text{MindFlayerSGD}}\left(\text{Med}[\eta]\right)} \geq \alpha.$$

*Proof.* Let $t = \text{Med}[\eta] =: m$, then we have

$$T_{\text{MindFlayerSGD}}(m) \leq \frac{K}{p} B(\tau + t) = 2KB\left(\tau + m\right),$$

$$T_{\text{RennalaSGD}} = KB(\tau + \mathbb{E}[\eta]) = KB(\tau + m + s),$$

Thus if $s > \tau + m$ then MindFlayer SGD is faster than Rennala SGD.

Now, let $s = (\tau + m)(2\alpha - 1)$ then

$$\frac{T_{\text{RennalaSGD}}}{T_{\text{MindFlayerSGD}}(m)} \geq \frac{\tau + m + s}{2(\tau + m)} = \frac{2\alpha(\tau + m)}{2(\tau + m)} = \alpha.$$

$\square$

# I   PROOFS FOR HOMOGENEOUS REGIME

## I.1   PROOF OF THEOREM 4.4

First, we rewrite MindFlayer SGD in a classical SGD way where we do gradient step with an unbiased estimator of the gradient at each iteration.

---

**Algorithm 9** MindFlayer SGD

1: **Input:** starting point $x^0$, stepsize $\gamma$, time budgets $t_1, \ldots, t_n \geq 0$, batch sizes $B_1, \ldots, B_n \geq 0$,
2: **for** $k = 0, 1, \ldots, K - 1$ **do**
3:    $g^k = \frac{1}{B} \sum_{i=1}^n \sum_{j=1}^{B_i} I\left(\eta_i^j \leq t_i\right) \nabla f\left(x^k; \xi_i^j\right)$
4:    $x^{k+1} = x^k - \gamma g^k$
5: **end for**

---

where $B = \sum_{i=1}^n p_i B_i$, $p_i = F(t_i) = P(\eta_i \leq t_i)$ and $I(\cdot)$ denotes the indicator function. To prove the theorem we need to establish some properties of the gradient estimator. First, we need an unbiased estimator.

**Lemma I.1** (Proof in Appendix I.1.1). *The gradient estimator in Algorithm 9 given by*

$$g(x) := \frac{1}{B} \sum_{i=1}^n \sum_{j=1}^{B_i} I\left(\eta_i^j \leq t_i\right) \nabla f\left(x; \xi_i^j\right)$$

*is unbiased, i.e., $\mathbb{E}[g(x)] = \nabla f(x)$ for all $x \in \mathbb{R}^d$.*

Next, we obtain an upper bound for the variance of this estimator.

**Lemma I.2** (Proof in Appendix I.1.2). *The gradient estimator in Algorithm 9 given by*

$$g(x) := \frac{1}{B} \sum_{i=1}^{n} \sum_{j=1}^{B_i} I\left(\eta_i^j \leq t_i\right) \nabla f\left(x; \xi_i^j\right)$$

*satisfies*

$$\mathbb{E}\left[\left\|g(x)^2\right\|\right] \leq 2 \left\|\nabla f(x)\right\|^2 + \frac{1}{B}\sigma^2.$$

We are ready to prove the Theorem 4.4.

**Theorem 4.4.** *Assume that Assumptions 4.1, 4.2 and 4.3 hold. Let $B = \sum_{i=1}^{n} p_i B_i$ and $\gamma = \frac{1}{2L} \min\left\{1, \frac{\varepsilon B}{\sigma^2}\right\}$ in Algorithm 1. Then, after*

$$K \geq \max\left\{1, \frac{\sigma^2}{\varepsilon B}\right\} \frac{8L\left(f(x^0) - f^{\inf}\right)}{\varepsilon}$$

*iterations, the method guarantees that $\frac{1}{K} \sum_{k=0}^{K-1} \mathbb{E}\left[\left\|\nabla f(x^k)\right\|^2\right] \leq \varepsilon$.*

*Proof.* Note that Algorithm 1 can be viewed as a special case of classical stochastic gradient descent (SGD), as reformulated in Algorithm 9. We need to verify that the gradient estimator fulfills the conditions required by classical SGD (Theorem G.2). The two preceding lemmas address this requirement precisely. Specifically, Lemma I.1 confirms that the gradient estimator used in Algorithm 9 is unbiased, while Lemma I.2 verifies that the variance of this estimator meets the conditions specified in Assumption G.1, with $\alpha = 0$, $\beta = 2$ and $\zeta = \frac{\sigma^2}{B}$. Consequently, it remains to apply Theorem G.2. $\square$

### I.1.1 PROOF OF LEMMA I.1

**Lemma I.1.** *The gradient estimator in Algorithm 9 given by*

$$g(x) := \frac{1}{B} \sum_{i=1}^{n} \sum_{j=1}^{B_i} I\left(\eta_i^j \leq t_i\right) \nabla f\left(x; \xi_i^j\right)$$

*is unbiased, i.e., $\mathbb{E}\left[g(x)\right] = \nabla f(x)$ for all $x \in \mathbb{R}^d$, where $B = \sum_{i=1}^{n} p_i B_i$.*

*Proof.* This follows from direct computation:

$$
\begin{aligned}
\mathbb{E}\left[g(x)\right] &= \mathbb{E}\left[\frac{1}{B} \sum_{i=1}^{n} \sum_{j=1}^{B_i} I\left(\eta_i^j \leq t_i\right) \nabla f\left(x; \xi_i^j\right)\right] \\
&= \frac{1}{B} \sum_{i=1}^{n} \sum_{j=1}^{B_i} \mathbb{E}\left[I\left(\eta_i^j \leq t_i\right) \nabla f\left(x; \xi_i^j\right)\right] \\
&\overset{\left(\eta_i^j \perp \xi_i^j\right)}{=} \frac{1}{B} \sum_{i=1}^{n} \sum_{j=1}^{B_i} \mathbb{E}\left[I\left(\eta_i^j \leq t_i\right)\right] \mathbb{E}\left[\nabla f\left(x; \xi_i^j\right)\right] \\
&= \frac{1}{B} \sum_{i=1}^{n} \sum_{j=1}^{B_i} p_i \nabla f(x) \\
&= \nabla f(x) \frac{1}{B} \sum_{i=1}^{n} p_i B_i \\
&= \nabla f(x).
\end{aligned}
$$

$\square$

### I.1.2 PROOF OF LEMMA I.2

**Lemma I.2.** *The gradient estimator in Algorithm 9 given by*

$$g(x) := \frac{1}{B} \sum_{i=1}^{n} \sum_{j=1}^{B_i} I\left(\eta_i^j \leq t_i\right) \nabla f\left(x; \xi_i^j\right)$$

*satisfies*

$$\mathbb{E}\left[\left\|g(x)^2\right\|\right] \leq 2 \left\|\nabla f(x)\right\|^2 + \frac{1}{B}\sigma^2,$$

*where $B = \sum_{i=1}^{n} p_i B_i$.*

*Proof.* In order to simplify notation, let

$$a_i := \sum_{j=1}^{B_i} b_i^j,$$

where

$$b_i^j := I\left(\eta_i^j \leq t_i\right) \nabla f\left(x; \xi_i^j\right).$$

**Step 1 (Initial expression).** We express $\mathbb{E}\left[\|g(x)\|^2\right]$ in terms of $a_i$:

$$\mathbb{E}\left[\|g(x)\|^2\right] = \mathbb{E}\left[\left\|\frac{1}{B} \sum_{i=1}^{n} a_i\right\|^2\right] = \frac{1}{B^2} \mathbb{E}\left[\sum_{i=1}^{n} \|a_i\|^2 + \sum_{i \neq j} \langle a_i, a_j \rangle\right].$$

We further simplify both terms via:

$$\|a_i\|^2 = \left\|\sum_{j=1}^{B_i} b_i^j\right\|^2 = \sum_{j=1}^{B_i} \left\|b_i^j\right\|^2 + \sum_{k \neq l} \langle b_i^k, b_i^l \rangle, \tag{7}$$

$$\langle a_i, a_j \rangle = \left\langle \sum_{k=1}^{B_i} b_i^k, \sum_{l=1}^{B_j} b_j^l \right\rangle = \sum_{k=1}^{B_i} \sum_{l=1}^{B_j} \langle b_i^k, b_j^l \rangle. \tag{8}$$

**Step 2. (Finding the expectations).** Further

$$\mathbb{E}\left[\left\|b_i^j\right\|^2\right] = \mathbb{E}\left[\left(I\left(\eta_i^j \leq t_i\right)\right)^2 \left\|\nabla f\left(x; \xi_i^j\right)\right\|^2\right]$$

$$\overset{(\eta_i^j \perp\!\!\!\perp \xi_i^j)}{=} \mathbb{E}\left[\left(I\left(\eta_i^j \leq t_i\right)\right)^2\right] \mathbb{E}\left[\left\|\nabla f\left(x; \xi_i^j\right)\right\|^2\right]$$

$$\leq p_i\left(\|\nabla f(x)\|^2 + \mathbb{E}\left[\left\|\nabla f\left(x; \xi_i^j\right) - \nabla f(x)\right\|^2\right]\right)$$

$$\overset{(Assumption\ 4.3)}{\leq} p_i\left(\|\nabla f(x)\|^2 + \sigma^2\right), \tag{9}$$

and

$$\mathbb{E}\left[\langle b_i^k, b_j^l \rangle\right] = \mathbb{E}\left[\langle I\left(\eta_i^k \leq t_i\right) \nabla f\left(x; \xi_i^k\right), I\left(\eta_j^l \leq t_j\right) \nabla f\left(x; \xi_j^l\right)\rangle\right]$$

$$\overset{(\perp\!\!\!\perp)}{=} \mathbb{E}\left[I\left(\eta_i^k \leq t_i\right)\right] \mathbb{E}\left[I\left(\eta_j^l \leq t_j\right)\right] \langle \mathbb{E}\left[\nabla f\left(x; \xi_i^k\right)\right], \mathbb{E}\left[\nabla f\left(x; \xi_j^l\right)\right]\rangle$$

$$= p_i p_j \|\nabla f(x)\|^2. \tag{10}$$

**Step 3 (Putting everything together).** We start with

$$\mathbb{E}\left[\|a_i\|^2\right] \overset{(7,9,10)}{\leq} B_i p_i\left(\|\nabla f(x)\|^2 + \sigma^2\right) + B_i\left(B_i - 1\right) p_i^2 \|\nabla f(x)\|^2$$

$$\leq B_i p_i\left(\|\nabla f(x)\|^2 + \sigma^2\right) + B_i^2 p_i^2 \|\nabla f(x)\|^2,$$

using this and recalling the definition of $B$, we get

$$\mathbb{E}\left[\sum_{i=1}^{n}\|a_i\|^2\right] \leq B\|\nabla f(x)\|^2 + B\sigma^2 + \|\nabla f(x)\|^2\sum_{i=1}^{n}B_i^2 p_i^2.$$

Next

$$\langle a_i, a_j \rangle \overset{(8,10)}{=} B_i p_i B_j p_j \|\nabla f(x)\|^2,$$

finally,

$$\mathbb{E}\left[\|g(x)\|^2\right] = \frac{1}{B^2}\mathbb{E}\left[\sum_{i=1}^{n}\|a_i\|^2 + \sum_{i\neq j}\langle a_i, a_j\rangle\right]$$

$$\leq \frac{1}{B^2}\left[B\|\nabla f(x)\|^2 + B\sigma^2 + \left(\sum_{i=1}^{n}B_i^2 p_i^2 + \sum_{i\neq j}B_i p_i B_j p_j\right)\|\nabla f(x)\|^2\right]$$

$$= \frac{1}{B^2}\left(B + B^2\right)\|\nabla f(x)\|^2 + \frac{\sigma^2}{B}$$

$$\leq 2\|\nabla f(x)\|^2 + \frac{\sigma^2}{B}.$$

$\square$

### I.2 PROOF OF THEOREM 4.5

The following lemma gives time complexity for any choice of $B_1, \ldots, B_n$ and $t = (t_1, \ldots, t_n)$ in MindFlayer SGD.

**Lemma I.3** (Proof in Appendix I.2.1)**.** *Assume that Assumptions 4.1, 4.2 and 4.3 hold. Let $B = \sum_{i=1}^{n}p_i B_i$ and $\gamma = \frac{1}{2L}\min\left\{1, \frac{\varepsilon B}{\sigma^2}\right\}$ in Method 1. Then after*

$$T_{\mathsf{MindFlayerSGD}}(t) \geq \max_{i\in[n]}\{B_i\left(\tau_i + t_i\right)\}\max\left\{1, \frac{\sigma^2}{\varepsilon B}\right\}\frac{8L\left(f(x_0) - f^{\inf}\right)}{\varepsilon}$$

*seconds, the method guarantees to find an $\epsilon$-stationary point.*

Now we are ready to prove the theorem.

**Theorem 4.5.** *Assume that Assumptions 4.1, 4.2 and 4.3 hold. Let $B = \sum_{i=1}^{n}p_i B_i$ and $\gamma = \frac{1}{2L}\min\left\{1, \frac{\varepsilon B}{\sigma^2}\right\}$ in Method 1. Let $t = (t_1, \ldots, t_n)$, $t_1, \ldots, t_n \geq 0$. Without loss of generality assume that $0 < \tau_1 + t_1 \leq \cdots \leq \tau_n + t_n$. Let*

$$t(m) = \left(\sum_{j=1}^{m}\frac{p_j}{\tau_j + t_j}\right)^{-1}\left(S + \sum_{j=1}^{m}p_j\right),$$

*where $S = \max\left\{1, \frac{\sigma^2}{\varepsilon}\right\}$. Let $m^* = \arg\min_{m\in[n]}t(m)$, if there are several minimizers we take the smallest one. Put*

$$B_i = \lceil b_i \rceil, \quad b_i = \begin{cases} \frac{t(m^*)}{\tau_i + t_i} - 1, & \text{if } i \leq m^*, \\ 0, & \text{if } i > m^*. \end{cases}$$

*Then, MindFlayer SGD guarantees to find an $\epsilon$-stationary point after*

$$T_{\mathsf{MindFlayerSGD}}(t) \geq 8 \times \min_{m\in[n]}\left\{\left(\frac{1}{m}\sum_{j=1}^{m}\frac{p_j}{\tau_j + t_j}\right)^{-1}\left(\frac{S}{m} + \frac{1}{m}\sum_{j=1}^{m}p_j\right)\frac{\Delta L}{\varepsilon}\right\}$$

*seconds, where $\Delta = f(x_0) - f^{\inf}$.*

*Proof.* First we show that $B_i$-s are valid choice, i.e. $b_i > 0$ for $i \leq m^*$. If $m^* = 1$, then $t(1) = \frac{\tau_1 + t_1}{p_1}(S + p_1)$, thus $b_1 = \frac{S}{p_1} > 0$. If $m^* > 1$, then, by its definition, $t(m^*) < t(m^* - 1)$. This implies

$$\left(\sum_{j=1}^{m^*} \frac{p_j}{\tau_j + t_j}\right)^{-1} \left(S + \sum_{j=1}^{m^*} p_j\right) < \left(\sum_{j=1}^{m^*-1} \frac{p_j}{\tau_j + t_j}\right)^{-1} \left(S + \sum_{j=1}^{m^*-1} p_j\right),$$

leading to

$$\left(\sum_{j=1}^{m^*-1} \frac{p_j}{\tau_j + t_j}\right) \left(S + \sum_{j=1}^{m^*} p_j\right) < \left(\sum_{j=1}^{m^*} \frac{p_j}{\tau_j + t_j}\right) \left(S + \sum_{j=1}^{m^*-1} p_j\right)$$

and

$$p_{m^*} \left(\sum_{j=1}^{m^*} \frac{p_j}{\tau_j + t_j}\right) < \frac{p_{m^*}}{\tau_{m^*} + t_{m^*}} \left(S + \sum_{j=1}^{m^*} p_j\right).$$

From the last inequality, we get that $\tau_{m^*} + t_{m^*} < t(m^*)$, thus $b_i \geq b_{m^*} > 0$ for all $i \leq m^*$.

It remains to find the time complexity with these choices of $B_i$. From Lemma I.3, we have that the time complexity is

$$T_{\text{MindFlayerSGD}}(t) \geq \max_{i \in [n]} \{B_i (\tau_i + t_i)\} \max \left\{1, \frac{\sigma^2}{\varepsilon B}\right\} \frac{8\Delta L}{\varepsilon}.$$

Then,

$$\max_{i \in [n]} \{B_i (\tau_i + t_i)\} \leq \max_{i \in [n]} \{(b_i + 1)(\tau_i + t_i)\} = t(m^*).$$

On the other hand

$$B = \sum_{i=1}^{n} p_i B_i \geq \sum_{i=1}^{n} p_i b_i = \sum_{i=1}^{m^*} t(m^*) \frac{p_i}{\tau_i + t_i} - \sum_{i=1}^{m^*} p_i$$

$$= \left(\sum_{j=1}^{m^*} \frac{p_j}{\tau_j + t_j}\right)^{-1} \left(S + \sum_{j=1}^{m^*} p_j\right) \sum_{i=1}^{m^*} \frac{p_i}{\tau_i + t_i} - \sum_{i=1}^{m^*} p_i = S \geq \frac{\sigma^2}{\varepsilon}.$$

Therefore, the time complexity is

$$T_{\text{MindFlayerSGD}}(t) \geq t(m^*) \frac{8\Delta L}{\varepsilon}$$

$$= \min_{m \in [n]} \left\{ \left(\sum_{j=1}^{m} \frac{p_j}{\tau_j + t_j}\right)^{-1} \left(S + \sum_{j=1}^{m} p_j\right) \right\} \frac{8\Delta L}{\varepsilon}.$$

$\square$

### I.2.1 PROOF OF LEMMA I.3

**Lemma I.3.** *Assume that Assumptions 4.1, 4.2 and 4.3 hold. Let $B = \sum_{i=1}^{n} p_i B_i$ and $\gamma = \frac{1}{2L} \min\left\{1, \frac{\varepsilon B}{\sigma^2}\right\}$ in Method 1. Then after*

$$T_{\text{MindFlayerSGD}}(t) \geq \max_{i \in [n]} \{B_i (\tau_i + t_i)\} \max \left\{1, \frac{\sigma^2}{\varepsilon B}\right\} \frac{8L\left(f(x_0) - f^{\text{inf}}\right)}{\varepsilon}$$

*seconds, the method guarantees to find an $\epsilon$-stationary point.*

*Proof.* Let $T_i^j(t_i)$ be the random time taken by client $i$ in the $j$-th attempt of calculating gradient estimator. We have

$$T_i^j(t_i) = \begin{cases} \tau_i + \eta_i^j, & \text{if } \eta_i^j \leq t_i, \\ \tau_i + t_i, & \text{if } \eta_i^j > t_i. \end{cases} \tag{11}$$

Thus, the random time taken for client $i$ to finish it's all $b_i$ trials is

$$\mathcal{T}_i(t_i) := \sum_{j=1}^{b_i} T_i^j(t_i) \leq b_i \left( \tau_i + t_i \right).   \tag{12}$$

Finally, let $\mathcal{T}$ be the random time required for one iteration of MindFlayer SGD. We get

$$\mathcal{T} = \max_{i \in [n]} \mathcal{T}_i(t_i) \leq \max_{i \in [n]} \{ b_i \left( \tau_i + t_i \right) \}.   \tag{13}$$

It remains to multiply $\mathcal{T}$ with the number of iterations $K$ given by Theorem 4.4. $\qquad\square$

## J    PROOFS FOR HETEROGENEOUS REGIME

### J.1    PROOF OF THEOREM D.1

Here, we rewrite Vecna SGD (Algorithm 3) in a classical SGD way.

---

**Algorithm 10** Vecna SGD

---

1: **Input:** starting point $x^0$, stepsize $\gamma$, time budgets $t_1, \ldots, t_n \geq 0$, batch sizes $b_1, \ldots, b_n \geq 0$,
2: **for** $k = 0, 1, \ldots, K - 1$ **do**
3: $\quad g^k = \frac{1}{n} \sum_{i=1}^n \frac{1}{p_i B_i} \sum_{j=1}^{B_i} I \left( \eta_i^j \leq t_i \right) \nabla f_i \left( x^k; \xi_i^j \right)$
4: $\quad x^{k+1} = x^k - \gamma g^k$
5: **end for**

---

where $p_i = F(t_i) = P(\eta_i \leq t_i)$.

To prove the theorem we need to establish some properties of the gradient estimator. First, we need an unbiased estimator.

**Lemma J.1** (Proof in Appendix J.1.1). *The gradient estimator in Algorithm 10 given by*

$$g(x) := \frac{1}{n} \sum_{i=1}^n \frac{1}{p_i B_i} \sum_{j=1}^{B_i} I \left( \eta_i^j \leq t_i \right) \nabla f_i \left( x; \xi_i^j \right)$$

*is unbiased, i.e., $\mathbb{E}\left[ g(x) \right] = \nabla f(x)$ for all $x \in \mathbb{R}^d$.*

Next, we obtain an upper bound for the variance of this estimator.

**Lemma J.2** (Proof in Appendix J.1.2). *The gradient estimator in Algorithm 10 given by*

$$g(x) := \frac{1}{n} \sum_{i=1}^n \frac{1}{p_i B_i} \sum_{j=1}^{B_i} I \left( \eta_i^j \leq t_i \right) \nabla f_i \left( x; \xi_i^j \right)$$

*satisfies*

$$\mathbb{E}\left[ \left\| g(x)^2 \right\| \right] \leq \frac{2 \left( f(x_0) - f^{\inf} \right) L}{n^2} \sum_{i=1}^n \frac{1 - p_i}{p_i B_i} + \| \nabla f(x) \|^2 + \frac{\sigma^2}{n^2} \sum_{i=1}^n \frac{1}{p_i B_i}.$$

We are ready to prove Theorem D.1. First, let us restate the theorem.

**Theorem D.1.** *Assume that Assumptions 4.1, 4.2 hold for the function $f$ and Assumption 4.3 holds for the function $f_i$ for all $i \in [n]$. Let $\gamma = \min \left\{ \frac{1}{\sqrt{L\alpha K}}, \frac{1}{L\beta}, \frac{\varepsilon}{2L\zeta} \right\}$ in Algorithm 3. Then after*

$$K \geq \frac{12 \Delta L}{\varepsilon} \max \left\{ \beta, \frac{12 \Delta \alpha}{\varepsilon}, \frac{2\zeta}{\varepsilon} \right\},$$

*iterations, the method guarantees that $\min_{0 \leq k \leq K} \mathbb{E}\left[ \left\| \nabla f(x^k) \right\|^2 \right] \leq \varepsilon$, where $\Delta = f(x_0) - f^{\inf}$ and*

$$\alpha = \frac{L}{n^2} \sum_{i=1}^n \frac{1 - p_i}{p_i B_i}, \quad \beta = 1, \quad \zeta = \frac{\sigma^2}{n^2} \sum_{i=1}^n \frac{1}{p_i B_i}.$$

*Proof.* Note that Algorithm 3 can be viewed as a special case of classical stochastic gradient descent (SGD), as reformulated in Algorithm 10. We need to verify that the gradient estimator fulfills the conditions required by classical SGD (Theorem G.2). The two preceding lemmas address this requirement precisely. Specifically, Lemma J.1 confirms that the gradient estimator used in Algorithm 10 is unbiased, while Lemma J.2 verifies that the variance of this estimator meets the conditions specified in Assumption G.1. Consequently, it remains to apply Theorem G.2. □

### J.1.1 PROOF OF LEMMA J.1

**Lemma J.1.1.** *The gradient estimator in Algorithm 10 given by*

$$g(x) := \frac{1}{n} \sum_{i=1}^{n} \frac{1}{p_i B_i} \sum_{j=1}^{B_i} I\left(\eta_i^j \leq t_i\right) \nabla f_i\left(x; \xi_i^j\right)$$

*is unbiased, i.e.,* $\mathbb{E}\left[g(x)\right] = \nabla f(x)$ *for all* $x \in \mathbb{R}^d$.

*Proof.* This follows from direct computation:

$$
\begin{aligned}
\mathbb{E}\left[g(x)\right] &= \mathbb{E}\left[\frac{1}{n} \sum_{i=1}^{n} \frac{1}{p_i B_i} \sum_{j=1}^{B_i} I\left(\eta_i^j \leq t_i\right) \nabla f_i\left(x; \xi_i^j\right)\right] \\
&= \frac{1}{n} \sum_{i=1}^{n} \frac{1}{p_i B_i} \sum_{j=1}^{B_i} \mathbb{E}\left[I\left(\eta_i^j \leq t_i\right) \nabla f_i\left(x; \xi_i^j\right)\right] \\
&\stackrel{\left(\eta_i^j \perp \xi_i^j\right)}{=} \frac{1}{n} \sum_{i=1}^{n} \frac{1}{p_i B_i} \sum_{j=1}^{B_i} \mathbb{E}\left[I\left(\eta_i^j \leq t_i\right)\right] \mathbb{E}\left[\nabla f_i\left(x; \xi_i^j\right)\right] \\
&= \frac{1}{n} \sum_{i=1}^{n} \frac{1}{p_i B_i} \sum_{j=1}^{B_i} p_i \nabla f_i(x) \\
&= \frac{1}{n} \sum_{i=1}^{n} \nabla f_i(x) \\
&= \nabla f(x).
\end{aligned}
$$

□

### J.1.2 PROOF OF LEMMA J.2

**Lemma J.2.** *The gradient estimator in Algorithm 10 given by*

$$g(x) := \frac{1}{n} \sum_{i=1}^{n} \frac{1}{p_i B_i} \sum_{j=1}^{B_i} I\left(\eta_i^j \leq t_i\right) \nabla f_i\left(x; \xi_i^j\right)$$

*satisfies*

$$\mathbb{E}\left[\left\|g(x)^2\right\|\right] \leq \frac{2L\left(f(x_0) - f^{\mathrm{inf}}\right)}{n^2} \sum_{i=1}^{n} \frac{1 - p_i}{p_i B_i} + \|\nabla f(x)\|^2 + \frac{\sigma^2}{n^2} \sum_{i=1}^{n} \frac{1}{p_i B_i}.$$

*Proof.* Since $\eta_i^j$ and $\xi_i^j$ are independent from each other for all $i \in [n]$ and $j$, we have

$$\mathrm{Var}\left(g(x)\right) = \frac{1}{n^2} \sum_{i=1}^{n} \frac{1}{p_i^2 B_i^2} \sum_{j=1}^{B_i} \mathrm{Var}\left(I\left(\eta_i^j \leq t_i\right) \nabla f_i\left(x; \xi_i^j\right)\right),$$

then we use the fact that

$$\mathrm{Var}\left(XY\right) = \mathrm{Var}\left(X\right) \mathrm{Var}\left(Y\right) + \mathrm{Var}\left(X\right) \mathbb{E}\left[Y\right]^2 + \mathrm{Var}\left(Y\right) \mathbb{E}\left[X\right]^2,$$

where $X$ and $Y$ are independent random variables. Hence, we obtain the following bound on the variance

$$\text{Var}\left(I\left(\eta_i^j \leq t_i\right) \nabla f_i\left(x; \xi_i^j\right)\right) \leq p_i(1-p_i)\sigma^2 + p_i(1-p_i)\|\nabla f_i(x)\|^2 + \sigma^2 p_i^2$$

$$= p_i\sigma^2 + p_i(1-p_i)\|\nabla f_i(x)\|^2.$$

As a result, the variance of $g(x)$ is bounded by

$$\text{Var}(g(x)) \leq \frac{1}{n^2}\sum_{i=1}^{n}\frac{1}{p_i^2 B_i^2}\sum_{j=1}^{B_i}\left(p_i\sigma^2 + p_i(1-p_i)\|\nabla f_i(x)\|^2\right)$$

$$= \frac{1}{n^2}\sum_{i=1}^{n}\frac{1}{p_i B_i}\left(\sigma^2 + (1-p_i)\|\nabla f_i(x)\|^2\right).$$

Finally

$$\mathbb{E}\left[\left\|g(x)^2\right\|\right] = \text{Var}(g(x)) + \|\mathbb{E}[g(x)]\|^2$$

$$\leq \|\nabla f(x)\|^2 + \frac{1}{n^2}\sum_{i=1}^{n}\frac{1-p_i}{p_i B_i}\|\nabla f_i(x)\|^2 + \frac{\sigma^2}{n^2}\sum_{i=1}^{n}\frac{1}{p_i B_i}.$$

Next we use $\|\nabla f_i(x)\|^2 \leq 2L\left(f(x_0) - f^{\inf}\right)$, thus

$$\mathbb{E}\left[\left\|g(x)^2\right\|\right] \leq \frac{2L\left(f(x_0) - f^{\inf}\right)}{n^2}\sum_{i=1}^{n}\frac{1-p_i}{p_i B_i} + \|\nabla f(x)\|^2 + \frac{\sigma^2}{n^2}\sum_{i=1}^{n}\frac{1}{p_i B_i}.$$

$\square$

## J.2 PROOF OF THEOREM D.2

The following lemma gives time complexity for any choice of $B_1, \ldots, B_n$ and $t = (t_1, \ldots, t_n)$ in Vecna SGD.

**Lemma J.3** (Proof in Appendix J.2.1). *Assume that Assumptions 4.1, 4.2 hold for the function $f$ and Assumption 4.3 holds for the function $f_i$ for all $i \in [n]$. Let $\gamma = \min\left\{\frac{1}{\sqrt{L\alpha K}}, \frac{1}{L}, \frac{\varepsilon}{2L\zeta}\right\}$ in Algorithm 3. Then after*

$$T_{\text{VecnaSGD}}(t) \geq \max_{i\in[n]}\{B_i(\tau_i + t_i)\}\frac{12\Delta L}{\varepsilon}\max\left\{1, \frac{12\Delta\alpha}{\varepsilon}, \frac{2\zeta}{\varepsilon}\right\}$$

*seconds, where the method guarantees to find an $\epsilon$-stationary point, where $\Delta = f(x_0) - f^{\inf}$ and*

$$\alpha = \frac{L}{n^2}\sum_{i=1}^{n}\frac{1-p_i}{p_i B_i}, \quad \zeta = \frac{\sigma^2}{n^2}\sum_{i=1}^{n}\frac{1}{p_i B_i}.$$

Now we are ready to prove the theorem.

**Theorem D.2.** *Assume that Assumptions 4.1, 4.2 hold for the function $f$ and Assumption 4.3 holds for the function $f_i$ for all $i \in [n]$. Let $\gamma = \min\left\{\frac{1}{\sqrt{L\alpha K}}, \frac{1}{L}, \frac{\varepsilon}{2L}\right\}$ in Algorithm 3, where*

$$\alpha = \frac{L}{n^2}\sum_{i=1}^{n}\frac{1-p_i}{p_i B_i}, \quad \zeta = \frac{\sigma^2}{n^2}\sum_{i=1}^{n}\frac{1}{p_i B_i}.$$

*Let $t = (t_1, \ldots, t_n)$, $t_1, \ldots, t_n \geq 0$. Without loss of generality assume that $0 < \tau_1 + t_1 \leq \cdots \leq \tau_n + t_n$. Let*

$$T = \tau_n + t_n + \left[\frac{1}{n}\sum_{i=1}^{n}\frac{\tau_i + t_i}{p_i}\right]\frac{\sigma^2}{n\varepsilon} + \left[\frac{1}{n}\sum_{i=1}^{n}\frac{1-p_i}{p_i}(\tau_i + t_i)\right]\frac{\Delta L}{n\varepsilon},$$

*where $\Delta = f(x_0) - f^{\mathrm{inf}}$. Put*

$$B_i = \lceil b_i \rceil, \quad b_i = \frac{T}{\tau_i + t_i}.$$

*Then, Vecna SGD guarantees to find an $\epsilon$-stationary point after*

$$T_{\mathsf{VecnaSGD}}(t) \geq 288 \times \frac{\Delta L}{\varepsilon} \left( \tau_n + t_n + \left[ \frac{1}{n} \sum_{i=1}^{n} \frac{\tau_i + t_i}{p_i} \right] \frac{\sigma^2}{n\varepsilon} + \left[ \frac{1}{n} \sum_{i=1}^{n} \frac{1 - p_i}{p_i} (\tau_i + t_i) \right] \frac{\Delta L}{n\varepsilon} \right)$$

*seconds.*

*Proof.* Since we have $b_i \geq 1$ for all $i \in [n]$, we get

$$\max_{i \in [n]} \{ B_i (\tau_i + t_i) \} \leq \max_{i \in [n]} \{ (b_i + 1)(\tau_i + t_i) \} \leq 2 \max_{i \in [n]} \{ b_i (\tau_i + t_i) \} = 2T.$$

It remains to apply Lemma J.3. We get

$$\frac{12\Delta\alpha}{\varepsilon} = \frac{12\Delta L}{\varepsilon n^2} \sum_{i=1}^{n} \frac{1 - p_i}{p_i B_i} \leq \frac{12\Delta L}{\varepsilon n^2} \sum_{i=1}^{n} \frac{1 - p_i}{p_i b_i}$$

$$= \frac{12\Delta L}{n\varepsilon} \frac{1}{T} \frac{1}{n} \sum_{i=1}^{n} \frac{1 - p_i}{p_i} (\tau_i + \eta_i) \leq 12,$$

and

$$\frac{2\zeta}{\varepsilon} = \frac{2\sigma^2}{\varepsilon n^2} \sum_{i=1}^{n} \frac{1}{p_i B_i} \leq \frac{2\sigma^2}{\varepsilon n^2} \sum_{i=1}^{n} \frac{1}{p_i b_i} \leq \frac{2\sigma^2}{n\varepsilon} \frac{1}{T} \frac{1}{n} \sum_{i=1}^{n} \frac{\tau_i + t_i}{p_i} \leq 2.$$

Finally, we get that Algorithm 3 returns a solution after

$$T_{\mathsf{MindFlayerSGD}}(t) \geq \max_{i \in [n]} \{ B_i (\tau_i + t_i) \} \frac{12\Delta L}{\varepsilon} \max \left\{ 1, \frac{12\Delta\alpha}{\varepsilon}, \frac{2\zeta}{\varepsilon} \right\}$$

$$\geq 288 \frac{\Delta L}{\varepsilon} T$$

$$\geq 288 \frac{\Delta L}{\varepsilon} \left( \tau_n + t_n + \left[ \frac{1}{n} \sum_{i=1}^{n} \frac{\tau_i + t_i}{p_i} \right] \frac{\sigma^2}{n\varepsilon} + \left[ \frac{1}{n} \sum_{i=1}^{n} \frac{1 - p_i}{p_i} (\tau_i + t_i) \right] \frac{\Delta L}{n\varepsilon} \right)$$

seconds. $\qquad\square$

### J.2.1 PROOF OF LEMMA J.3

**Lemma J.3.** *Assume that Assumptions 4.1, 4.2 hold for the function $f$ and Assumption 4.3 holds for the function $f_i$ for all $i \in [n]$. Let $\gamma = \min\left\{ \frac{1}{\sqrt{L\alpha K}}, \frac{1}{L}, \frac{\varepsilon}{2L\zeta} \right\}$ in Algorithm 3. Then after*

$$T_{\mathsf{VecnaSGD}}(t) \geq \max_{i \in [n]} \{ B_i (\tau_i + t_i) \} \frac{12\Delta L}{\varepsilon} \max \left\{ 1, \frac{12\Delta\alpha}{\varepsilon}, \frac{2\zeta}{\varepsilon} \right\}$$

*seconds, where the method guarantees to find an $\epsilon$-stationary point, where $\Delta = f(x_0) - f^{\mathrm{inf}}$ and*

$$\alpha = \frac{L}{n^2} \sum_{i=1}^{n} \frac{1 - p_i}{p_i B_i}, \quad \zeta = \frac{\sigma^2}{n^2} \sum_{i=1}^{n} \frac{1}{p_i B_i}.$$

*Proof.* Let $T_i^j(t_i)$ be the random time taken by client $i$ in the $j$-th attempt of calculating gradient estimator. We have

$$T_i^j(t_i) = \begin{cases} \tau_i + \eta_i^j, & \text{if } \eta_i^j \leq t_i, \\ \tau_i + t_i, & \text{if } \eta_i^j > t_i. \end{cases} \tag{14}$$

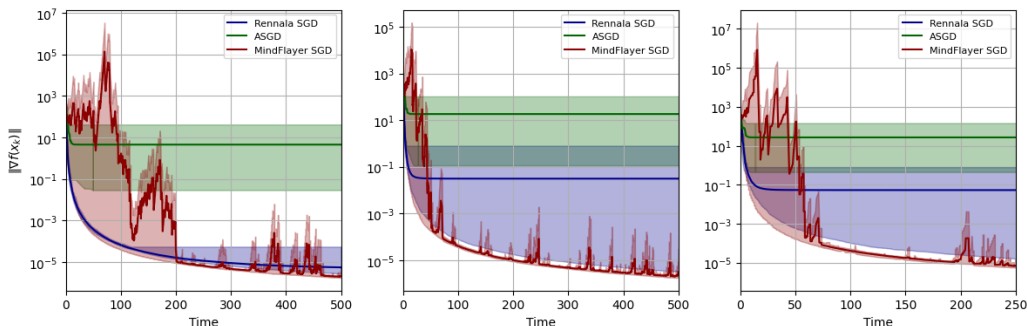

Figure 5: We ran an experiment as described in Section 6 where we employ the same $\mathcal{J}_i = $ InfBernoulli($q$) distribution for all clients $i \in [n]$, with different $q$ values. From left to right we have $q = 0.6, 0.7, 0.8$. Additionally, we set $\tau_i = \sqrt{i+1}$. As we observe, with an increase of the probability of failure $q$ unlike Rennala SGD and ASGD, MindFlayer SGD demonstrates the ability to continue optimizing and not be stuck

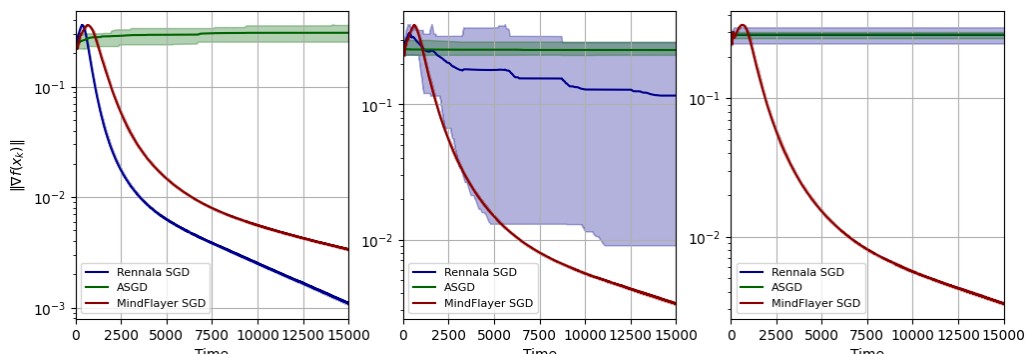

Figure 6: We train a two layer Neural Network on the MNIST dataset where we set the distribution $\mathcal{J}_i = $ Log-Cauchy($s$) for all clients $i \in [n]$, with different scale values $s$. From left to right we have $s = 1, 10, 100$. Additionally, we set $\tau_i = \sqrt{i+1}$. We observe that Mindflayer SGD convergence doesn't suffer from the increase in the scale parameter $s$. On the other hand, Rennala and ASGD are delayed significantly with bigger scale parameters $s$

Thus, the random time taken for client $i$ to finish it's all $B_i$ trials is

$$\mathcal{T}_i(t_i) := \sum_{j=1}^{B_i} T_i^j(t_i) \leq B_i (\tau_i + t_i). \tag{15}$$

Finally, let $\mathcal{T}$ be the random time required for one iteration of Vecna SGD. We get

$$\mathcal{T} = \max_{i \in [n]} \mathcal{T}_i(t_i) \leq \max_{i \in [n]} \{B_i (\tau_i + t_i)\}. \tag{16}$$

It remains to multiply $\mathcal{T}$ with the number of iterations $K$ given by Theorem D.1. □

## K    SUPPLEMENTAL FIGURES

