# OpenReview forum: "MindFlayer: Efficient Asynchronous Parallel SGD in the Presence of Heterogeneous and Random Worker Compute Times"
_ICLR.cc/2025/Conference — Submitted to ICLR 2025_

### Official Review · Reviewer_JaDR · 2024-10-31

**Soundness:** 3
**Presentation:** 3
**Contribution:** 2
**Rating:** 5
**Confidence:** 3

**Summary:**

This paper proposes a parallel SGD computation scheme when each local work has random compute times. The authors show that the proposed method is a generalization of Rennala SGD and is thus optimal in deterministic worker time regime. When randomness exists, the authors empirically verify that the proposed method is better than Rennala SGD and ASGD for different settings. Theory contribution involves single device justification of superiority of MindFlayer, derived convergence result and time complexity of MindFlayer.

**Strengths:**

This paper first time studies parallel SGD setting where workers have random reply times. This setting is practically important. Presentation is clear and easy to follow, with motivating single device example. Theorems are core (convergence and time complexity), accompanied with digestions. Experiments are illustrative and effectively demonstrate the superiority of the proposed method over baselines.

**Weaknesses:**

I personally recognize the main value of current work to be the setting being considered, where worker has random reply time. The authors claim they are first to consider this type of setting, and I do think it is of practical importance. The authors may consider addressing the following limitations:

1. The proposed method seems theoretically intuitive but may have practical limitations. Algorithm 1 and Algorithm 2 look more like an experiment setup instead of a true practical algorithm. For example, distribution of $\eta_i$ and thus also $p_i$ are hard to evaluate in practice. Compared to ASGD and Rennala SGD, Algorithm 1 involves more parameters such as $B_i,p_i$ that users need to decide on. Experiments are carried out where $\eta_i$  is of certain ideal theoretic distribution, which is rarely encountered in practice. Therefore, empirically estimation of good choices of $B_i,p_i$ and even $\mathcal{J}_i$ remain a problem.

2. The setting being considered is interesting but may rise other practical issues. For example, the proposed method chooses to initialize another query whenever the current gradient computation takes too lone (quantified by $>t$), I feel this would significantly increase number of communication rounds between server and workers compared to baseline ASGD and Rennala SGD, which may form bottleneck in certain settings.

3. Figure 3 is hard to read, legends should be added to demonstrate what are dashed green line/ dotted purple line/ solid yellow line/ shaded grey region. The title includes no such information as well.

Minor writing typos:
1. line 33, shouldn't the codomain of $f$ be $\mathbb{R}$ instead of $\mathbb{R}^d$?
2. line 5 in Algorithm 1 and line 308 of text, the authors seem to intend to refer Algorithm 2 while wrongly typed Algorithm 4? If Algorithm 4 is intended, it should be moved to main content.
3. line 346, it says "convergence of Rennala SGD" while I feel it's for  MindFlayer instead of Rennala SGD?
4. in Figure 3 title, "section" missed i
5. line 464, "the time complexity of Rennala SGD" should capitalize first letter

**Questions:**

see limitations above

---

> ### Author Response · Authors · 2024-11-25
> **Response to the Weaknesses**
>
> > 1. The proposed method seems theoretically intuitive but may have practical limitations. Algorithm 1 and Algorithm 2 look more like an experiment setup instead of a true practical algorithm. For example, distribution of $\eta_i$ and thus also $p_i$ are hard to evaluate in practice. Compared to ASGD and Rennala SGD, Algorithm 1 involves more parameters such as $B_i, p_i$ that users need to decide on. Experiments are carried out where $\eta_i$ is of certain ideal theoretic distribution, which is rarely encountered in practice. Therefore, empirically estimation of good choices of $B_i, p_i$ and even $\mathcal{J}_i$ remain a problem.
>
> To address these concerns, we have included a new section titled "Simplifying MindFlayer for Practical Use" in the appendix, which introduces Mod MindFlayer SGD, a more practical variant. This simplified version reduces the reliance on precise distribution knowledge by replacing $t_i$​ with the inverse cumulative distribution function (CDF) at a probabilistic threshold $p$. The parameter $t_i$​ can be estimated using historical data or dynamically updated via the Robbins-Monro stochastic approximation, making the approach feasible even when exact distributions are unavailable. Additionally, global parameters $p$ and $B$ replace per-worker parameters $t_i$​ and $B_i$​, significantly reducing the tuning burden for users.
>
> While the original algorithms aim to provide theoretical insights, Mod MindFlayer SGD bridges the gap between theory and practice, addressing concerns about parameter estimation and real-world applicability without sacrificing performance, as demonstrated by our empirical results.
>
> > 2. The setting being considered is interesting but may rise other practical issues. For example, the proposed method chooses to initialize another query whenever the current gradient computation takes too lone (quantified by >t), I feel this would significantly increase number of communication rounds between server and workers compared to baseline ASGD and Rennala SGD, which may form bottleneck in certain settings.
>
> It depends on how the method is implemented. The most efficient approach would be to avoid adding extra communication steps by sending the value of tit_iti​ to each worker at the start. Since each worker knows the time they are allotted to compute a gradient, they can restart the computation themselves if it takes too long, without needing to communicate with the server.
>
> Even if communication were necessary, it could be as minimal as a single bit of information, making it unlikely to form a significant bottleneck.
>
> > 3. Figure 3 is hard to read, legends should be added to demonstrate what are dashed green line/ dotted purple line/ solid yellow line/ shaded grey region. The title includes no such information as well.
>
> Sorry about that, a figure with a legend is included in the revision.
>
> ---
>
> Thank you for pointing out the typos, they have been addressed in the revision.

---

### Official Review · Reviewer_Gd16 · 2024-11-03

**Soundness:** 3
**Presentation:** 3
**Contribution:** 2
**Rating:** 5
**Confidence:** 3

**Summary:**

This paper proposed a new asynchronous SGD method for minimizing smooth and nonconvex functions, called MindFlayer SGD, which is specifically designed to operate under conditions of heterogeneous and random compute times. Through theoretical analysis and empirical evaluation, the authors show that MindFlayer SGD outperforms existing methods, including Rennala SGD, especially in scenarios with heavy-tailed noise. In addition, the paper also introduced Vecna SGD in the heterogeneous regime with applications in the distributed optimization and federated learning

**Strengths:**

1. **Well-Organized Structure**: The paper is structured clearly, allowing readers to follow the development of ideas seamlessly. It includes a comprehensive literature review that contextualizes the research within the broader field of stochastic optimization, highlighting the significance of addressing the challenges posed by heterogeneous and random compute times. The authors present the MindFlayer SGD algorithm with detailed explanations of its design and motivation. This clarity enhances the reader's understanding of how the algorithm functions and its advantages over existing methods.

2. **Strong Supporting Evidence**: The claims made in the paper are robustly supported by both theoretical proofs and empirical results. The theoretical framework provides a solid foundation for the proposed algorithm, while the experimental findings demonstrate its superior performance in various scenarios, particularly in the presence of heavy-tailed noise.

3. **Sufficient Implementation and Proof Details**: The paper provides adequate details regarding the implementation of MindFlayer SGD and the associated proofs. This transparency allows for reproducibility and facilitates further research, enabling other researchers to build on the authors' findings effectively.

**Weaknesses:**

**Implicit Assumption of Utilizing Privileged Information $p_i$ in Algorithm Design**. In the real world, the distributions of compute time for different works are usually unknown. The proposed algorithms implicitly leverage this information to get the expected batch size $B$. I am not sure if this is required in the proof. The paper fails to discuss this limitation and provide a workaround without using information $p_i$

**Questions:**

## Major
1. Implicit Use of Privileged Information  $p_i$: Could the authors clarify this assumption further? Why is it necessary? Is there a possibility of incorporating an online estimate of $p_i$ within the algorithm?
2. Decentralized Setting: The paper primarily addresses a centralized setting. Are there any potential extensions to accommodate decentralized training?


## Minor
1. Method 4 in Algorithm 1 should be referred to as Algorithm 2
2. Algorithms 2 and 4 could be misleading. I don't think we can realistically sample compute times in the real world.

---

> ### Author Response · Authors · 2024-11-25
> **Response to Weaknesses**
>
> > **Implicit Assumption of Utilizing Privileged Information in Algorithm Design**. In the real world, the distributions of compute time for different works are usually unknown. The proposed algorithms implicitly leverage this information to get the expected batch size $B$. I am not sure if this is required in the proof. The paper fails to discuss this limitation and provide a workaround without using information $p_i$
>
> We acknowledge the implicit assumption of leveraging privileged information, such as worker compute time distributions, in the proposed algorithms. To address this, we added a section titled "Simplifying MindFlayer for Practical Use" in the appendix, introducing Mod MindFlayer SGD, which reduces reliance on precise distribution knowledge.
>
> Mod MindFlayer SGD replaces $t_i$​ with the inverse cumulative distribution function (CDF) at a probabilistic threshold $p$, estimated using historical data or online updates via the Robbins-Monro stochastic approximation. This reformulation simplifies implementation by dynamically adapting $t_i$​​ based on observed compute times, aligning it with the desired reliability $p$.
>
> Experimental results show that Mod MindFlayer SGD achieves comparable performance to the original algorithm while addressing this limitation and enhancing practicality for real-world scenarios.

---

> > ### Author Response · Authors · 2024-11-25
> > **Response to the Questions**
> >
> > Major:
> > > 1. Implicit Use of Privileged Information $p_i$: Could the authors clarify this assumption further? Why is it necessary? Is there a possibility of incorporating an online estimate of $p_i$ within the algorithm?
> >
> > Addressed above.
> >
> > > 2. Decentralized Setting: The paper primarily addresses a centralized setting. Are there any potential extensions to accommodate decentralized training?
> >
> > We believe that before exploring the decentralized setting, it is essential to fully understand the centralized setting. There is still significant work to be done in this area, such as addressing communication complexity.
> >
> > ---
> >
> > Minor:
> >
> > > 1. Method 4 in Algorithm 1 should be referred to as Algorithm 2
> >
> > Thank you, this has been addressed in the revision.
> >
> > > 2. Algorithms 2 and 4 could be misleading. I don't think we can realistically sample compute times in the real world.
> >
> > We do not sample the actual computation time itself. Instead, we only care whether this random computation time is smaller than the allotted time $t$. In other words, if a worker finishes computing its gradient within the time frame $t$, we never need to know the exact computation time. Thank you for pointing this out—we understand that this point might be confusing, and we will add a clarification on this in the revision.

---

### Official Review · Reviewer_kAF6 · 2024-11-03

**Soundness:** 2
**Presentation:** 3
**Contribution:** 2
**Rating:** 3
**Confidence:** 4

**Summary:**

The paper explores the challenges and solutions of SGD in environments where compute times of parallel workers are irregular and unpredictable. Prior work introduced Rennala SGD for situations with heterogeneous but fixed compute times, optimizing time complexity. However, this method falters when applied to scenarios where worker compute times vary randomly. The authors introduce MindFlayer SGD, a new asynchronous SGD method designed to handle random compute times and demonstrate its effectiveness through both theoretical analysis and empirical data, showing that it outperforms existing methods like Rennala SGD, especially in environments with heavy-tailed noise distributions.

**Strengths:**

1. MindFlayer SGD provides a more realistic approach to asynchronous SGD in modern, heterogeneous computing environments.
2. The paper provides a solid theoretical framework for the proposed method, including proofs of its efficiency and effectiveness over traditional methods

**Weaknesses:**

1. As described in Algortihms 1 and 2, a client needs to wait for other clients even after completing all of its trials, and each server’s update (Algorithm 1 Line 12) must aggregates gradients from all clients. Given these characteristics, MindFlayer SGD seems to function more like a synchronous algorithm rather than an asynchronous one.
2. The generalized computation model modestly extends the fixed computation model used in Rennala SGD, which may not present significant technical challenges.
3. The impact of various hyperparameters involved in MindFlayer SGD is not deeply discussed Particularly, the parameters $t_i$ and $B_i$ are difficult to choose in practice, which plays a central role in MindFlayer SGD and could be crucial for practitioners aiming to adapt the method to specific applications.
4. The experiments compare MindFlayer SGD solely with Rennala SGD and vanilla ASGD, and the test problems may be overly simple. Including comparisons with a wider array of contemporary asynchronous methods on more complex machine learning problems could enhance the argument for MindFlayer SGD's superiority.

**Questions:**

1. In line 470, th authors use L-BFGS-B to obtain optimal $t$. How to obtain the unknown parameters, e.g., $p_j$, in the optimization problem? Is the optimality theoretically guaranteed? What is the computational cost incurred?
2. In Algorithm 1 Line 5, Line 307, and Line 323, shoud the “Method 4” or “Algorithm 4” be changed to ”Algorithm 2“?

---

> ### Author Response · Authors · 2024-11-25
> **Response to the Weaknesses**
>
> > 1. As described in Algortihms 1 and 2, a client needs to wait for other clients even after completing all of its trials, and each server’s update (Algorithm 1 Line 12) must aggregates gradients from all clients. Given these characteristics, MindFlayer SGD seems to function more like a synchronous algorithm rather than an asynchronous one.
>
> You are right; our algorithm is not fully asynchronous but instead partially synchronous, as each client performs a varying number of gradient steps. This isn't a drawback—in fact, it's an advantage. Just like the optimal Rennala algorithm in deterministic time scenarios, ours is not fully asynchronous either. We have even benchmarked it against the fully asynchronous ASGD algorithm and found that our MindFlayer algorithm performs better.
>
> We agree that the word "asynchronous" in the title may sound confusing, and we are considering removing it during the revision.
>
> The issue with ASGD is that its workers operate independently, completing their calculations of stochastic gradients with potentially large and unpredictable delays. They apply their results as soon as they are ready, without waiting for other workers. This approach is problematic, as it slows down convergence when the stochastic gradient is calculated based on an outdated or irrelevant point. When the computation times of workers are random, these points can become very outdated and chaotic, which only exacerbates the problem.
>
> > 2. The generalized computation model modestly extends the fixed computation model used in Rennala SGD, which may not present significant technical challenges.
>
> To the best of our knowledge, we are the first to consider this generalized computation model, where deterministic computation times are replaced with random ones. While this may appear to be a modest extension, it represents a significant technical step forward. Unlike previous work, we make no assumptions about the distributions governing this randomness, allowing for a much broader applicability of the model.
>
> Furthermore, we developed a method that not only recovers Rennala SGD in the deterministic setting but also outperforms it in scenarios where the random computation times exhibit heavy-tailed behavior. This demonstrates the robustness and adaptability of our approach to more complex and realistic settings.
>
> > 3. The impact of various hyperparameters involved in MindFlayer SGD is not deeply discussed. Particularly, the parameters $t_i$ and $B_i$ are difficult to choose in practice, which plays a central role in MindFlayer SGD and could be crucial for practitioners aiming to adapt the method to specific applications.
>
> We have added a section titled "Simplifying MindFlayer for Practical Use" to the appendix, introducing Mod MindFlayer SGD, a streamlined variant that replaces $t_i$ and $B_i$​ with two global parameters: a probabilistic threshold $p$ and a global batch size $B$. This reformulation simplifies implementation while retaining the core principles and robustness of the original algorithm.
>
> To facilitate $t_i$​ selection, we propose a dynamic adjustment using the Robbins-Monro stochastic approximation, aligning $t_i$​ with the desired completion probability $p$. Experimental results in the appendix demonstrate that Mod MindFlayer SGD performs comparably to the original algorithm, highlighting that this simplification preserves the essential behavior of MindFlayer SGD while making it more practical for real-world applications.
>
> > 4. The experiments compare MindFlayer SGD solely with Rennala SGD and vanilla ASGD, and the test problems may be overly simple. Including comparisons with a wider array of contemporary asynchronous methods on more complex machine learning problems could enhance the argument for MindFlayer SGD's superiority.
>
> The algorithms included in our comparisons were chosen for their theoretical significance. ASGD serves as a well-established baseline, and Rennala is optimal under fixed compute time, making them particularly relevant for evaluating MindFlayer SGD. For this reason, our analysis focuses on these algorithms to provide a clear and theoretically grounded comparison.

---

> > ### Author Response · Authors · 2024-11-25
> > **Response to the Questions**
> >
> > > 1. In line 470, th authors use L-BFGS-B to obtain optimal $t$. How to obtain the unknown parameters, e.g., $p_j$, in the optimization problem? Is the optimality theoretically guaranteed? What is the computational cost incurred?
> >
> > The section in question serves to compare the algorithms empirically, as a full theoretical comparison is intractable due to the complexities involved. Specifically, for any choice of distributions $\mathcal{J}_i$​, all parameters except the optimal $t$ are known, making it effectively an $n$-dimensional optimization problem in $\mathbb{R}^n$. To address this, we use the L-BFGS-B algorithm, which is well-suited for solving smooth and mildly nonconvex optimization problems, to obtain optimal $t$. This provides a benchmark against which we compare our heuristic approach of using the median.
> >
> > > 2. In Algorithm 1 Line 5, Line 307, and Line 323, shoud the "Method 4" or "Algorithm 4" be changed to "Algorithm 2"?
> >
> > Thank you, this has been addressed in the revision.

---

### Official Review · Reviewer_f3wW · 2024-11-06

**Soundness:** 3
**Presentation:** 1
**Contribution:** 2
**Rating:** 3
**Confidence:** 3

**Summary:**

This paper proposes MindFlayer, a variant of asynchronous SGD that specifically deal with the presence of random compute times. Theoretical analysis and empirical results are provided accordingly.

**Strengths:**

1. This paper proposes MindFlayer, a variant of asynchronous SGD that specifically deal with the presence of random compute times.

2. Theoretical analysis and empirical results are provided accordingly.

**Weaknesses:**

Major issues:

1. For the algorithm itself, the contribution of MindFlayer SGD is incremental since it's mostly based on Rennala SGD, as the authors stated in this paper by themselves: "For MindFlayer SGD, each iteration, on average, receives only Bp gradients, making it effectively a scaled-down version of Rennala SGD." (Lin 241-242)

2. Although it is stated that MindFlayer SGD is designed for dealing with the complexities/challenges of real-world distributed learning environments and applications, the experiment settings are far from those for real-world applications (especially in 2024). The only non-convex problem used in the experiment is an extremely simple two-layer neural network on the MNSIT dataset, which is far from convincing if the authors want to show that the proposed algorithm could be used in real-world distributed training and applications.
Regardless of the real-world settings, most of the experiments are on convex problems, which cannot even be used to verify the theoretical results, since the theoretical analysis is based on smooth but non-convex settings.

3. I cannot find the hyperparameters (basically just learning rates) used in the experiments, or in what range the hyperparameters are tuned. Actually, it is known whether the baselines are fairly tuned. For example, in Figure 5, the curve of ASGD seems to be a flat line (or it simply gets worse from the very beginning). However, if the learning rate is well tuned, a small enough learning rate should make the grad norm or loss of ASGD going down even a little bit. Thus, the overall experiment settings and the hyperparameter tuning is questionable.

Minor issues:

1. In several algorithm, for example Algorithm 1, does "Method 4" actually mean Algorithm 4 in the appendix? If so, please make the naming consistent across the entire paper.
2. Separating the main algorithm (Algorithm 1 and Algorithm 4) in the main paper and the appendix is not friendly to the readers. Especially there is still space remaining in the main paper.
3. Putting experiment description and the corresponding figures (Figure 4,5) separately in main paper and the appendix is also unfriendly to the readers.

**Questions:**

1. How are the hyperparameters tuned in the experiments?

2. For non-convex problems, smaller gradient norm doesn't always implies smaller loss value. And eventually we train models for smaller loss, not for smaller gradient norms. Could the loss curve also be added for the experiment of the neural network on MNIST?

---

> ### Author Response · Authors · 2024-11-25
> **Response to the Weaknesses**
>
> Major issues:
>
> > 1. For the algorithm itself, the contribution of MindFlayer SGD is incremental since it's mostly based on Rennala SGD, as the authors stated in this paper by themselves: "For MindFlayer SGD, each iteration, on average, receives only Bp gradients, making it effectively a scaled-down version of Rennala SGD." (Lin 241-242)
>
> While we acknowledge that MindFlayer SGD builds upon Rennala SGD, we respectfully emphasize that this advancement carries significant theoretical and practical implications.
>
> First, the transition from fixed times to general random times is a substantial theoretical leap. The fact that a method, that generalizes Rennala in the fixed-time setting, performs robustly in the random-time setting highlights the strength and utility of our approach. This generalization is not merely an adaptation; it is a meaningful extension that advances the understanding of stochastic optimization under broader assumptions.
>
> Second, science inherently progresses through incremental improvements that stand on the foundation of previous work. The true measure of a contribution lies in the value of these refinements. Our work demonstrates that even small changes to Rennala SGD can lead to significant, measurable, and theoretically grounded performance gains. Importantly, as we illustrate in Figures 1, 3, 4, and 5, any improvement factor is, in principle, possible with our method. This highlights that MindFlayer SGD can, under the right conditions, outperform Rennala SGD arbitrarily.
>
>
> > 2. Although it is stated that MindFlayer SGD is designed for dealing with the complexities/challenges of real-world distributed learning environments and applications, the experiment settings are far from those for real-world applications (especially in 2024). The only non-convex problem used in the experiment is an extremely simple two-layer neural network on the MNSIT dataset, which is far from convincing if the authors want to show that the proposed algorithm could be used in real-world distributed training and applications. Regardless of the real-world settings, most of the experiments are on convex problems, which cannot even be used to verify the theoretical results, since the theoretical analysis is based on smooth but non-convex settings.
>
> Our primary goal is to validate and illustrate the theoretical contributions of MindFlayer SGD, not to address specific real-world problems. The simplicity of our experimental setup is intentional and aligns with the theoretical focus of our work. Following Occam's razor, we believe that simpler experiments often provide clearer insights than unnecessarily complex setups. Our experiments are carefully designed to demonstrate the behavior of our method and its theoretical guarantees in a controlled and interpretable manner.
>
> Conducting experiments on large-scale problems requires significant computational resources. Given the theoretical nature of this work, we see no need to simulate artificially large-scale systems. Our main contributions stand independently of such experiments.
>
> Do you agree that our experiments effectively corroborate our theoretical findings?
>
>
> > 3. I cannot find the hyperparameters (basically just learning rates) used in the experiments, or in what range the hyperparameters are tuned. Actually, it is known whether the baselines are fairly tuned. For example, in Figure 5, the curve of ASGD seems to be a flat line (or it simply gets worse from the very beginning). However, if the learning rate is well tuned, a small enough learning rate should make the grad norm or loss of ASGD going down even a little bit. Thus, the overall experiment settings and the hyperparameter tuning is questionable.
>
> Thank you for your observation regarding the absence of step size details in the original submission. We apologize for this omission and appreciate your diligence in bringing it to our attention. For most of the experiments, we followed the same step size and hyperparameter tuning approach as described in the Rennala paper. This has now been clarified in the revised version of our manuscript.
>
> Regarding the performance of ASGD, as depicted in Figure 5, the flat or worsening trend in the grad norm is an artifact of the timescale used for visualization. The scale of random times essentially stretches out the curve which explains the upward trend observed in ASGD values.
>
> ---
>
> Minor issues:
>
> We have resolved the issue with the algorithm numbering; please see the updated version on OpenReview.
> We also acknowledge the concern that separating the experiment descriptions in the main paper from the corresponding figures in the appendix is not reader-friendly. This decision was made to save space, but we will address this issue in the camera-ready version of the paper.

---

> > ### Author Response · Authors · 2024-11-25
> > **Response to the Questions**
> >
> > > 1. How are the hyperparameters tuned in the experiments?
> >
> > Addressed above.
> >
> > > 2. For non-convex problems, smaller gradient norm doesn't always implies smaller loss value. And eventually we train models for smaller loss, not for smaller gradient norms. Could the loss curve also be added for the experiment of the neural network on MNIST?
> >
> > In the optimization literature for first-order methods, theoretical guarantees are almost always formulated in terms of the gradient norm rather than the loss. This is because first-order methods can converge to stationary points, where the gradient is zero, and the method ceases to move further, remaining in the neighborhood of the stationary point. Importantly, these stationary points may not correspond to the minimum value of the loss, making it impossible to establish guarantees directly in terms of the loss function.
> >
> > We have provided theoretical convergence guarantees based on the gradient norm, as detailed in Section F (The Classical SGD Theory) of the appendix, and have referenced foundational works in the field that adopt a similar approach. For this reason, our analysis and experiments focus on the gradient norm, which is the standard metric for evaluating convergence in non-convex settings.

---

### Author Response · Authors · 2024-11-25
**New version of the paper**

Dear Reviewers,

Thank you for your reviews. We will address all of your concerns and questions individually soon.

In the meantime, we have uploaded a new version of our paper with the following changes:

- We addressed some minor issues mentioned by the reviewers, such as typos, reference/naming inconsistencies, and figure labeling errors.

- We added a new section, Section E, titled *"Simplifying MindFlayer for Practical Use"*. In this section, we propose \algname{Mod MindFlayer SGD}, a practical variant of our algorithm. This version replaces $B_i$ and $t_i$ with two global parameters: a probabilistic threshold $p$, which reflects the likelihood of completing a gradient computation, and a global batch size $B$, specifying the total number of trials across all workers. This reformulation simplifies hyperparameter tuning while retaining robustness.

- We updated the list of contributions in Section 2 to include:
  > In Appendix E, we present a simple modification of our algorithm, \algname{Rennala SGD}, which we call \algname{Mod MindFlayer SGD}. This version is more suitable for practical implementation.

---

### Meta-Review · Area_Chair_kkiT · 2024-12-19

**Metareview:**

The paper introduces MindFlayer SGD, an asynchronous SGD method for environments with random compute times, claiming better performance than Rennala SGD, especially with heavy-tailed distributions.  While the paper addresses a realistic scenario and provides a theoretically grounded algorithm with empirical support, its contributions are incremental over other algorithms. Further, the method relies on potentially unavailable information about compute time distributions and the experimental validation is limited in scope, using simplistic problems and few baselines. Practical concerns about hyperparameter tuning and a modified, more practical version of the algorithm are also noted. Due to these limitations and the need for more rigorous validation, the paper is recommended for rejection.

**Additional Comments On Reviewer Discussion:**

Unfortunately, the reviewers did not interact with the authors during the discussion phase, but it seems that it would have been difficult to drastically change their opinion.

---

### Decision · Program_Chairs · 2025-01-22

Reject